# DuaRot: Dual Rotation for Advanced Outlier Mitigation in Rotated LLMs

## Abstract

By employing rotation, outliers in activations can be effectively mitigated without altering the output, thereby facilitating the quantization of large language models (LLMs). However, existing rotation-based methods only consider global activation distributions, leaving the finer-grained distributions underexplored. Additionally, these methods predominantly rely on the Walsh–Hadamard transform (WHT) to accelerate online rotation operations, while not fully considering performance between matrix multiplication (Matmul) and WHT in actual runtime. These limitations hinder the rotation's ability to effectively reduce quantization errors and decrease inference speed. Therefore, improvements are needed in their performance regarding both accuracy and speed. In this paper, we propose a dual rotation method for rotation matrices, dubbed DuaRot, based on reparameterization. During training, DuaRot sequentially refines global and local features to achieve effective outlier mitigation. During inference, global and local rotations can be merged, which maintains rotational invariance without introducing additional computational overhead. Meanwhile, we propose a hardware-aware matrix configuration strategy, which determines whether the online Hadamard matrix should be expanded into a trainable parameter space by taking the runtime of the WHT and Matmul into account. This approach further enhances the reduction of quantization errors in online rotation operations without compromising inference speed. Extensive experiments demonstrate that DuaRot outperforms existing methods across various models and quantization configurations. For instance, when applied to LLaMA3-8B, DuaRot achieves WikiText-2 perplexities of 7.49 and 7.41 under W4A4KV4 and W4A4KV16 configurations with Round-to-Nearest (RTN), improving by 0.51 and 0.41 over the state-of-the-art, respectively. The code will be publicly available soon.

## 1 Introduction

In recent years, Large Language Models (LLMs) (Floridi & Chiriatti, 2020; Jiang et al., 2023; AI@Meta, 2024; Yang et al., 2024) have rapidly emerged, demonstrating remarkable effectiveness across various fields (Zellers et al., 2019; Hendrycks et al., 2020; Zhang et al., 2023; Wu et al., 2023; Mo et al., 2024). Nevertheless, the performance of LLMs heavily relies on a large number of parameters, which always leads to significant memory, computational overhead and high energy consumption during deployment. To reduce overhead while retaining performance for LLMs, it is vital to study the network compression. Among many compression methods (Yao et al., 2022; Frantar & Alistarh, 2023; Lin et al., 2024a), model quantization has garnered significant attention from both industry and academia. The goal of model quantization is to convert high-precision weights/activations to low-precision and replace high-precision operations with low-precision ones, thereby reducing the memory footprint and computational resources needed deploying these models.

However, the presence of outliers in activations often leads to a significant accuracy drop when directly quantizing LLMs. Although LLM.int8() (Dettmers et al., 2022) separates outliers and uses mixed-precision matrix multiplication to minimize quantization errors, this fine-grained approach often decreases the model's inference speed. As one of the most representative works in LLM quantization, SmoothQuant (Xiao et al., 2023) handles outliers through scale invariance. It shifts the

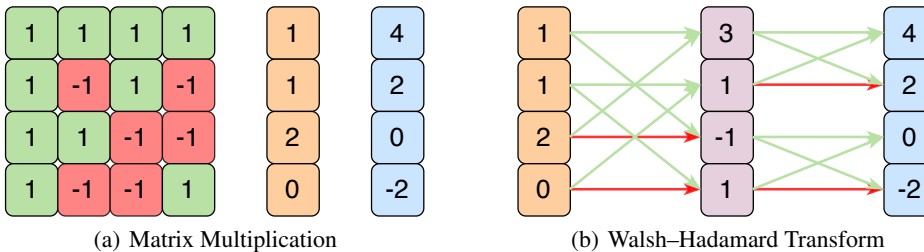

(a) Matrix Multiplication           (b) Walsh–Hadamard Transform

Figure 1: Comparison between Matmul and WHT. Unlike Matmul, which obtains results through Multiply-Add ($4 = 1 \times 1 + 1 \times 1 + 1 \times 2 + 1 \times 0$), WHT uses reduction to perform Add and Sub ($4 = 3 + 1 = (1 + 2) + (1 + 0)$) to get the results. Green and red represent +1 and -1, respectively. Best viewd in colors.

quantization challenge from activations to weights, thereby reducing quantization errors for activations and enhancing network performance. Subsequent research has further improved the effectiveness of scaling-based methods by employing layer-wise search (Wei et al., 2023) or introducing trainable parameters (Shao et al., 2023). Although these methods achieve improvement for 6-bit and 8-bit activation quantization, all of them fail when activations are quantized to 4-bit.

Recently, the rotation-based method QuaRot (Liu et al., 2024) has attracted significant attention from the community. By leveraging rotational invariance (Ashkboos et al., 2024a), QuaRot effectively disperses outliers across channels, reducing the activation outliers and enhancing quantization performance. Building on this, SpinQuant (Liu et al., 2024) further optimizes the rotation matrices using Cayley optimization (Li et al., 2020) to enhance rotation quality. While QuaRot and SpinQuant enhance 4-bit activation quantization performance, they neglect fine-grained feature distribution and need to be further improved. Additionally, they depend on the Walsh–Hadamard transform (WHT) to simplify matrix multiplication (Matmul) and accelerate online operations. This approach fails to take into account the superior matrix operation capabilities of GPUs. Naively applying WHT not only reduces inference speed but may also compromise effectiveness of rotation matrix. For example, SpinQuant's peak performance relies on GPTQ (Frantar et al., 2022), yet it often underperforms when using Round-to-Nearest (RTN). Meanwhile, both QuaRot and SpinQuant slow down inference speed for decoding stage.

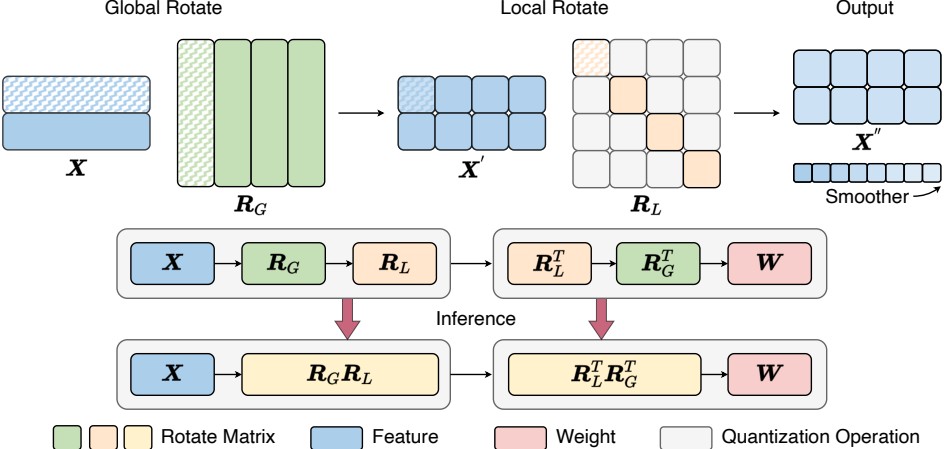

Figure 2: The framework of DuaRot. During training, DuaRot achieves a smoother distribution by sequentially applying global and local rotations to the activations. During inference, the properties of the rotation matrices enable merging the two matrices according to Eq 9 and Eq 10, thereby maintaining rotational invariance without introducing any additional computational overhead.

In this paper, our goal is to further refine the activation distribution and enhance the quantization accuracy of LLMs without sacrificing inference speed. We propose a reparameterization strategy for rotation matrix, termed *Dual Rotation* (DuaRot). Specifically, as shown in Figure 2, we utilize global rotation matrix to capture broader patterns, while local rotation matrix target specific anomalies. Incorporating both global and local rotation matrix improves their capacity to detect and mitigate out-

liers. During inference, these two rotation matrices can be merged, maintaining rotational invariance and introducing no additional computational burdens. Additionally, we develop a hardware-aware matrix configuration strategy. By comparing the runtime between WHT and Matmul for matrices of various sizes in prefill, low-pressure decoding, and high-pressure decoding scenarios, we determine whether to extend the online Hadamard matrix into trainable space. This approach enhances their ability to mitigate outliers for online operations, and improves model accuracy without compromising inference speed. Benefit from these, DuaRot exhibits significant improvements over previous methods across various quantization techniques, particularly under RTN.

Our contributions are summarized as follows:

- We introduce a novel reparameterization strategy for rotation matrices called DuaRot, which incorporates both global and local rotations to enhance adaptability during training. This dual-layered approach enhances the ability of rotation matrices to detect and eliminate outliers, thereby improving model accuracy. After training, these rotation matrices can be efficiently merged into a single matrix without introducing additional overhead during inference.

- We propose a hardware-aware matrix configuration strategy that determines the trainability of online rotation operations based on runtime comparisons between WHT and Matmul, rather than relying solely on offline computational feasibility. This approach further reduces quantization errors of activations without sacrificing inference speed.

- DuaRot demonstrates superior accuracy and computational efficiency compared to existing approaches, especially in the context of RTN quantization. Compared to SpinQuant, our approach improves perplexity (PPL) on the WikiText-2 dataset by 0.32, 0.24, 0.51, and 0.12 for the W4A4KV4 configuration, and by 0.31, 0.21, 0.41, and 0.15 for the W4A4KV16 configuration for LLaMA2-7B, LLaMA2-13B, LLaMA3-8B, and Mistral-7B, respectively.

## 2 RELATED WORK

Eliminating outliers is crucial for reducing quantization errors and improving accuracy for LLM quantization. Currently, outlier elimination methods can be primarily categorized into two types: scaling-based methods and rotation-based methods.

### 2.1 ELIMINATING OUTLIERS THROUGH SCALING

SmoothQuant (Xiao et al., 2023) is one of the most representative methods in LLM quantization, being the first to propose using scale invariance to transfer outliers from activation to weights. This approach reduces quantization error and enhances the network's performance in W8A8. Outlier Suppression+ (Wei et al., 2023) identifies that outliers tend to cluster in specific channels and exhibit asymmetry across different channels. It proposes channel-wise shifting and scaling to address this issue, migrating these operations into subsequent modules to maintain equivalence. Furthermore, OmniQuant (Shao et al., 2023) introduces learnable weight clipping and equivalent transformations, operating within a differentiable framework through block-wise error minimization. QQQ (Zhang et al., 2024) proposes adaptive smoothing, which achieves higher-quality activation quantization and improved accuracy. However, scaling-based methods merely transfer the quantization challenge from activations to weights, failing to fundamentally address the overall outlier problem. These methods can complicate the quantization of weights, particularly when outliers in activation are exceptionally large (Sun et al., 2024). Thus, how to effectively and efficiently address the outlier problem in LLM quantization still remains a significant challenge.

### 2.2 ELIMINATING OUTLIERS THROUGH ROTATION

Recent research has shown that employing rotation matrices can effectively mitigate outliers in LLMs. A pioneering work in this field is QuIP (Chee et al., 2024), which suggests that incoherence processing serves as a technique for mitigating outliers in both the weight and activation spaces. QuIP enhances the incoherence of weight and Hessian matrices by multiplying them with a random orthogonal matrix generated using the Kronecker product. Subsequently, QuIP# (Tseng et al., 2024) employs a randomized Hadamard transform, which is faster and demonstrates better theoretical properties than previous QuIP. Building on prior research, QuaRot (Ashkboos et al., 2024b) is

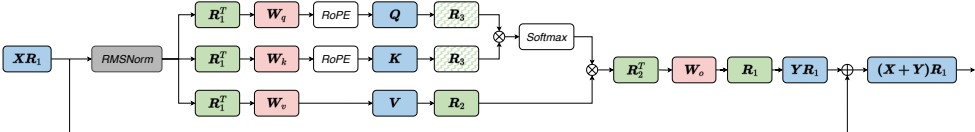

(a) Orthogonal transformations for Multi-Head Attention (MHA)

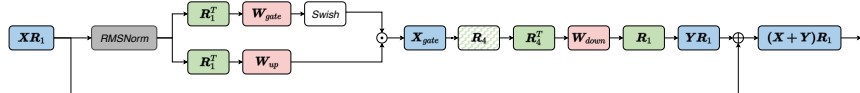

(b) Orthogonal transformations for Feed-Forward Network (FFN)

Figure 3: An illustration for rotational invariance. Inserting a rotation matrix and its transpose between the activation values and weights, results in invariance in the computation results and effectively eliminates outliers in the activation values. Inserting a rotation matrix $\boldsymbol{R}_1$ before the residual connections of the MHA and FFN results in consistent rotation for the inputs of the following layer: $\boldsymbol{X}\boldsymbol{R}_1 + \boldsymbol{Y}\boldsymbol{R}_1 = (\boldsymbol{X} + \boldsymbol{Y})\boldsymbol{R}_1$. $(\boldsymbol{R}_1, \boldsymbol{R}_1^T, \boldsymbol{R}_2, \boldsymbol{R}_2^T, \boldsymbol{R}_4^T)$ can be absorbed into the weight offline and $(\boldsymbol{R}_3, \boldsymbol{R}_4)$ should be online computed.

the first to connect the properties of rotation matrices for outlier mitigation with invariance transformation. By applying rotation matrices to LLaMA (Touvron et al., 2023), QuaRot significantly improves the quantization performance of post-training quantization (PTQ). Additionally, it finds that the randomized Hadamard transform offers a significant improvement over the random orthogonal transform. SpinQuant Liu et al. (2024) extends the rotation matrices into the trainable parameter space and employs Cayley optimization (Li et al., 2020) to optimize these matrices, further enhancing their performance. QServe (Lin et al., 2024b) is the first study to integrate both scaling and rotation techniques. It eliminates outliers in the inputs to the MHA and FFN blocks through randomized Hadamard transforms, while also employing scaling techniques to remove outliers within the blocks, thus improving performance on W4A8KV4.

## 3 METHOD

### 3.1 ROTATIONAL INVARIANCE IN LLMS

In this section, we introduce rotational invariance (Ashkboos et al., 2024a) in LLMs. Without loss of generality, as shown in Figure 3, we use the structure of LLaMA as an example. We fuse the scaling factor $\boldsymbol{\alpha}$ of RMSNorm into the weights through constant folding, and omit the $\boldsymbol{\alpha}$ from our subsequent discussions in this paper for simplicity.

Each LLaMA decoder layer consists of a Multi-Head Attention (MHA) and a Feed-Forward Network (FFN) and both of which utilize pre-norm (Xiong et al., 2020). Benefit from the commutation property that $\text{RMSNorm}(\boldsymbol{X}\boldsymbol{R}_1) = \text{RMSNorm}(\boldsymbol{X})\boldsymbol{R}_1$ (Ashkboos et al., 2024a), where we assume here that $\boldsymbol{R}_1\boldsymbol{R}_1^T = \boldsymbol{I}$ and RMSNorm is applied to each row of the activations $\boldsymbol{X}$ as $\boldsymbol{x}_i \leftarrow \boldsymbol{x}_i/\|\boldsymbol{x}_i\|$, if we have consistently $\boldsymbol{Y}\boldsymbol{R}_1 = F(\boldsymbol{X})\boldsymbol{R}_1$ and wish to get function $F'$ where $\boldsymbol{Y}\boldsymbol{R}_1 = F'(\boldsymbol{X}\boldsymbol{R}_1)$, it is equivalent to transform the $(\boldsymbol{W}_q, \boldsymbol{W}_k, \boldsymbol{W}_v, \boldsymbol{W}_o)$ to $(\boldsymbol{R}_1^T\boldsymbol{W}_q, \boldsymbol{R}_1^T\boldsymbol{W}_k, \boldsymbol{R}_1^T\boldsymbol{W}_v, \boldsymbol{W}_o\boldsymbol{R}_1)$ in MHA and transform the $(\boldsymbol{W}_{gate}, \boldsymbol{W}_{up}, \boldsymbol{W}_{down})$ to $(\boldsymbol{R}_1^T\boldsymbol{W}_{gate}, \boldsymbol{R}_1^T\boldsymbol{W}_{up}, \boldsymbol{W}_{down}\boldsymbol{R}_1)$ in FFN:

$$\text{MHA}(\boldsymbol{X}|\boldsymbol{W}_q, \boldsymbol{W}_k, \boldsymbol{W}_v, \boldsymbol{W}_o)\boldsymbol{R}_1 = \text{MHA}(\boldsymbol{X}\boldsymbol{R}_1|\boldsymbol{R}_1^T\boldsymbol{W}_q, \boldsymbol{R}_1^T\boldsymbol{W}_k, \boldsymbol{R}_1^T\boldsymbol{W}_v, \boldsymbol{W}_o\boldsymbol{R}_1) \quad (1)$$

$$\text{FFN}(\boldsymbol{X}|\boldsymbol{W}_{gate}, \boldsymbol{W}_{up}, \boldsymbol{W}_{down})\boldsymbol{R}_1 = \text{FFN}(\boldsymbol{X}\boldsymbol{R}_1|\boldsymbol{R}_1^T\boldsymbol{W}_{gate}, \boldsymbol{R}_1^T\boldsymbol{W}_{up}, \boldsymbol{W}_{down}\boldsymbol{R}_1). \quad (2)$$

According to the distributive law of matrix multiplication, we can infer that the output of the residual connection will also multiply by $\boldsymbol{R}_1$:

$$\boldsymbol{X}\boldsymbol{R}_1 + \text{MHA}(\boldsymbol{X}\boldsymbol{R}_1|\boldsymbol{R}_1^T\boldsymbol{W}_q, \boldsymbol{R}_1^T\boldsymbol{W}_k, \boldsymbol{R}_1^T\boldsymbol{W}_v, \boldsymbol{W}_o\boldsymbol{R}_1) = \boldsymbol{X}\boldsymbol{R}_1 + \boldsymbol{Y}\boldsymbol{R}_1 = (\boldsymbol{X} + \boldsymbol{Y})\boldsymbol{R}_1 \quad (3)$$

$$\boldsymbol{X}\boldsymbol{R}_1 + \text{FFN}(\boldsymbol{X}\boldsymbol{R}_1|\boldsymbol{R}_1^T\boldsymbol{W}_{gate}, \boldsymbol{R}_1^T\boldsymbol{W}_{up}, \boldsymbol{W}_{down}\boldsymbol{R}_1) = \boldsymbol{X}\boldsymbol{R}_1 + \boldsymbol{Y}\boldsymbol{R}_1 = (\boldsymbol{X} + \boldsymbol{Y})\boldsymbol{R}_1 \quad (4)$$

On the other hand, from the perspective of the inner workings of MHA and FFN, by inserting the head-wise rotation matrices $\boldsymbol{R}_2$ and $\boldsymbol{R}_2^T$ between $\boldsymbol{W}_v$ and $\boldsymbol{W}_o$, as well as inserting a pair of $\boldsymbol{R}_3$ between $\boldsymbol{Q}$ and $\boldsymbol{K}$ after RoPE, and $\boldsymbol{R}_4$ and $\boldsymbol{R}_4^T$ before $\boldsymbol{W}_{down}$, we can achieve rotational invariance:

$$\boldsymbol{A}\boldsymbol{V}\boldsymbol{R}_2\boldsymbol{R}_2^T\boldsymbol{W}_o = \boldsymbol{A}\boldsymbol{V}\boldsymbol{W}_o, \boldsymbol{Q}\boldsymbol{R}_3(\boldsymbol{K}\boldsymbol{R}_3)^T = \boldsymbol{Q}\boldsymbol{K}^T, \boldsymbol{X}_{gate}\boldsymbol{R}_4\boldsymbol{R}_4^T\boldsymbol{W}_{down} = \boldsymbol{X}_{gate}\boldsymbol{W}_{down}, \quad (5)$$

where $\boldsymbol{A}$ is attention matrix.

Based on this, transforming $(\boldsymbol{W}_{embedding}, \boldsymbol{W}_{lm\_head})$ to $(\boldsymbol{W}_{embedding}\boldsymbol{R}_1, \boldsymbol{R}_1^T \boldsymbol{W}_{lm\_head})$ allows that the input will change from $\boldsymbol{X}\boldsymbol{W}_{embedding}$ to $\boldsymbol{X}\boldsymbol{W}_{embedding}\boldsymbol{R}_1$ and the final output of the network remains unchanged:

$$\begin{aligned}
\text{LLaMA}(\boldsymbol{X}\boldsymbol{W}_{embedding})\boldsymbol{W}_{lm\_head} &= \boldsymbol{Y}\boldsymbol{W}_{lm\_head} \\
&= (\boldsymbol{Y}\boldsymbol{R}_1)\boldsymbol{R}_1^T \boldsymbol{W}_{lm\_head} = \text{LLaMA}(\boldsymbol{X}\boldsymbol{W}_{embedding}\boldsymbol{R}_1)\boldsymbol{R}_1^T \boldsymbol{W}_{lm\_head}
\end{aligned} \tag{6}$$

In other words, we have applied a rotational invariance transformation to the network.

## 3.2 DUAL ROTATION

In recent years, reparameterization (Ding et al., 2021) has been proven to be an effective technique and has been widely applied in various fields of computer vision, achieving significant performance improvements. During training, reparameterization introduces multiple branches to capture diverse features and enhance representational capacity. During inference, these branches are mathematically merged into a single equivalent layer, simplifying the model to achieve VGG-like efficiency while retaining the learned performance.

Inspired by reparameterization techniques, we propose *Dual Rotation* (DuaRot), a novel reparameterization method for rotation which aims at enhancing both the accuracy and efficiency of models. DuaRot involves a global rotation matrix $\boldsymbol{R}_G \in \mathbb{R}^{d\times d}$ and a local rotation matrix $\boldsymbol{R}_L \in \mathbb{R}^{d\times d}$, which contains $n$ groups. $\boldsymbol{R}_L$ is a block diagonal matrix:

$$\boldsymbol{R}_L = \text{BlockDiag}(\mathbf{R}_L), \text{where } \mathbf{R}_L \in \mathbb{R}^{n\times \frac{d}{n}\times \frac{d}{n}} \text{ and } (\mathbf{R}_L)_i(\mathbf{R}_L)_i^T = \boldsymbol{I}, \forall i \tag{7}$$

During training, we utilize both $\boldsymbol{R}_G$ and $\boldsymbol{R}_L$ to capture diverse activation distribution. Specifically, as shown in Figure 2, $\boldsymbol{R}_G$ encompasses a global rotation that applies a comprehensive rotation across the entire dimensional space. This enables the model to learn broad and holistic transformations effectively. Meanwhile, $\boldsymbol{R}_L$ operates on a finer granular level, allowing for local rotations within smaller subspaces of the original dimensional space:

$$\boldsymbol{X}\boldsymbol{R}_G\boldsymbol{R}_L = \left[(\boldsymbol{X}^1\boldsymbol{R}_G)_{:,:\frac{d}{n}}(\mathbf{R}_L)_1; (\boldsymbol{X}^1\boldsymbol{R}_G)_{:,\frac{d}{n}:\frac{2d}{n}}(\mathbf{R}_L)_2; ...; (\boldsymbol{X}^1\boldsymbol{R}_G)_{:,\frac{(n-1)d}{n}:d}(\mathbf{R}_L)_n\right] \tag{8}$$

This bifocal approach helps reduce quantization errors and eliminate outliers by applying fine-tuned adjustments where necessary, thereby enhancing the accuracy of the model.

For the inference, as $\boldsymbol{X}\boldsymbol{R}_G\boldsymbol{R}_L = \boldsymbol{X}(\boldsymbol{R}_G\boldsymbol{R}_L)$, matrix can be mathematically combined and represented as $\boldsymbol{R} = \boldsymbol{R}_G\boldsymbol{R}_L$. Furthermore, from Eq 7, we can know $\boldsymbol{R}_L$ is an orthogonal matrix:

$$\boldsymbol{R}_L\boldsymbol{R}_L^T = \begin{bmatrix} (\mathbf{R}_L)_1(\mathbf{R}_L)_1^T & 0 & \cdots & 0 \\ 0 & (\mathbf{R}_L)_2(\mathbf{R}_L)_2^T & \cdots & 0 \\ \vdots & \vdots & \ddots & \vdots \\ 0 & 0 & \cdots & (\mathbf{R}_L)_n(\mathbf{R}_L)_n^T \end{bmatrix} = \begin{bmatrix} \boldsymbol{I} & 0 & \cdots & 0 \\ 0 & \boldsymbol{I} & 0 \\ \vdots & \vdots & \ddots & \vdots \\ 0 & 0 & \cdots & \boldsymbol{I} \end{bmatrix} \tag{9}$$

Therefore, $\boldsymbol{R}$ is also orthogonal:

$$\boldsymbol{R}\boldsymbol{R}^T = (\boldsymbol{R}_G\boldsymbol{R}_L)(\boldsymbol{R}_G\boldsymbol{R}_L)^T = \boldsymbol{R}_G\boldsymbol{R}_L\boldsymbol{R}_L^T\boldsymbol{R}_G^T = \boldsymbol{R}_G\boldsymbol{R}_G^T = \boldsymbol{I} \tag{10}$$

By merging $\boldsymbol{R}_G$ and $\boldsymbol{R}_L$, we can maintain the rotational invariance of the LLM without introducing any additional computational burden to the network.

## 3.3 HARDWARE-AWARE MATRIX CONFIGURATION STRATEGY

Previous works (Tseng et al., 2024; Ashkboos et al., 2024b; Liu et al., 2024) have demonstrated that Hadamard matrices can achieve faster and more accurate experimental results with superior theoretical properties. However, although Hadamard matrices of size $2^n$ use WHT to compute the vector-matrix product $\boldsymbol{x}\boldsymbol{H}$ in $\mathcal{O}(d\log_2(d))$, this method only considers the computational complexity and does not take into account the runtime efficiency.

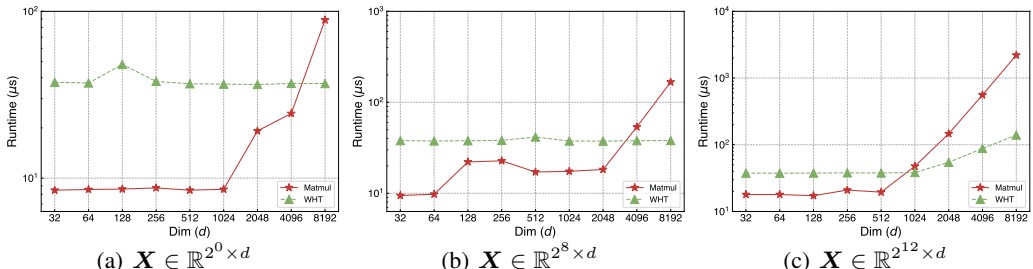

Figure 4: The runtime comparison of the WHT and Matmul for the computation of $\boldsymbol{XH}$ on an NVIDIA A100-SXM4-80GB under the different settings of $\boldsymbol{X}$ and $\boldsymbol{H} \in \mathbb{R}^{d \times d}$. We performed computations for $\boldsymbol{XH}$ using torch.float16 and measured the average time over 1000 runs using torch.utils.benchmark.

Compared to WHT, modern GPUs are highly optimized for Matmul and can further accelerate it through techniques such as blocking or packing. These methods enable Matmul to be more efficient than WHT in some cases, despite its higher computational complexity. Motivated by this, we measure the performance of the WHT and Matmul for the computation of $\boldsymbol{XH}$ on an NVIDIA A100-SXM4-80GB for $\boldsymbol{X} \in \mathbb{R}^{L \times d}$, where $L \in \{2^0, 2^8, 2^{12}\}$ to simulate the low-pressure decoding, high-pressure decoding, and prefill stages and $d \in \{2^6, ..., 2^{14}\}$. As seen in Figure 4, when the dimension is less than 512, the performance of Matmul is significantly higher than that of the WHT. However, when the dimension exceeds 512, the advantage of the WHT in reducing computational complexity begins to manifest as the computation scales.

In this paper, we propose a hardware-aware configuration strategy for rotation matrices that determines whether a rotation matrix is trainable based on its size and hardware runtime, rather than merely considering whether it can be computed offline:

$$\boldsymbol{R}^{d \times d} = \begin{cases} \text{Trainable,} & \boldsymbol{R} \text{ is offline,} \\ \text{Trainable,} & \boldsymbol{R} \text{ is online and } d \leq 512, \\ \text{Hadamard matrix,} & \boldsymbol{R} \text{ is online and } d > 512. \end{cases} \tag{11}$$

By employing this method, we can further reduce the quantization error associated with Hadamard transformations, thereby enhancing the model's accuracy without sacrificing inference speed.

## 4 EXPERIMENTS

### 4.1 EXPERIMENTAL SETTINGS

We conduct extensive experiments on the LLaMA models (2-7B/13B (Touvron et al., 2023) and 3-8B (AI@Meta, 2024)) and the Mistral-7B (Jiang et al., 2023) for DuaRot. We evaluate PPL on WikiText-2 for Language Generation Tasks. Meanwhile, we report the accuracy on eight zero-shot common sense reasoning tasks, including PIQA (Bisk et al., 2020), WinoGrande (Sakaguchi et al., 2021), HellaSwag (Zellers et al., 2019), ARC-challenge and ARC-easy (Clark et al., 2018), OBQA (Mihaylov et al., 2018), BoolQ (Clark et al., 2019) and SIQA (Sap et al., 2019). lm_eval==0.4.3 (Gao et al., 2024) are adopted with default parameters.

We employ Cayley SGD (Li et al., 2020) to optimize $\boldsymbol{R}_1$, $\boldsymbol{R}_2$, $\boldsymbol{R}_3$ and $\boldsymbol{R}_4$ while keep their orthogonality. All the weights in the networks are kept as constants. $\boldsymbol{R}_1 \in \mathbb{R}^{D_{token} \times D_{token}}$ is seen as global rotational matrix $\boldsymbol{R}_G$ and we employ a local rotational matrix $\boldsymbol{R}_L \in \mathbb{R}^{\frac{D_{token}}{d} \times d \times d}$, where $d$ is set to 64 to enhance accuracy of models. To inherit the excellent initialization properties of $\boldsymbol{R}_1$, we initialize $\boldsymbol{R}_L^i$ as $\boldsymbol{I}$. During inference, $\boldsymbol{R}_L$ can be fused into $\boldsymbol{R}_1$ ($\boldsymbol{R}_G$) as mentioned above and without introducing additional inference burden. $\boldsymbol{R}_2 \in \mathbb{R}^{N \times D_{head} \times D_{head}}$ and $\boldsymbol{R}_3 \in \mathbb{R}^{N \times D_{head} \times D_{head}}$ are head-wise rotational matrix to eliminate outliers in Value and Query-Key, respectively. Both of them are initialized as randomized Hadamard matrix and separately learned for each head. For $\boldsymbol{R}_4 \in \mathbb{R}^{m \times m}$, where $m$ is intermediate_size, we separate it into two Hadamard matrices via Kronecker product $\boldsymbol{R}_4 = \boldsymbol{R}_4^1 \otimes \boldsymbol{R}_4^2$ and determine whether to extend them to trainable space via Eq 11.

Table 1: Comparison of the WikiText-2 perplexity (↓) results for LLaMA and Mistral. The 4-4-4, 4-4-16 and 4-8-4 represent W4A4KV4, W4A4KV16 and W4A8KV4, respectively. We show the failed GPTQ experiments using NaN and the perplexity results>100 by Inf. Results for RTN, GPTQ and QuaRot (Ashkboos et al., 2024b) are obtained using QuaRot publicly released codebase. Results for SpinQuant (Liu et al., 2024) are obtained using their publicly released codebase. More quantization results, including W4A4KV8, W4A8KV8, W4A8KV16 are in the Appendix.

| Method | LLaMA2-7B | | | LLaMA2-13B | | | LLaMA3-8B | | | Mistral-7B | | |
|---|---|---|---|---|---|---|---|---|---|---|---|---|
| Baseline | 5.47 | | | 4.88 | | | 6.13 | | | 5.25 | | |
| | 4-4-4 | 4-4-16 | 4-8-4 | 4-4-4 | 4-4-16 | 4-8-4 | 4-4-4 | 4-4-16 | 4-8-4 | 4-4-4 | 4-4-16 | 4-8-4 |
| RTN | NaN | NaN | 7.92 | Inf | Inf | 5.79 | Inf | Inf | 19.38 | Inf | Inf | 12.70 |
| +QuaRot | 9.04 | 8.69 | 6.89 | 6.31 | 6.23 | 5.51 | 11.06 | 10.47 | 7.81 | 6.26 | 6.19 | 5.70 |
| +SpinQuant | 6.20 | 6.17 | 5.56 | 5.51 | 5.40 | 4.97 | 8.00 | 7.82 | 6.79 | 5.60 | 5.58 | 5.28 |
| +DuaRot | 5.88 | 5.86 | 5.51 | 5.27 | 5.19 | 4.94 | 7.49 | 7.41 | 6.76 | 5.48 | 5.43 | 5.24 |
| GPTQ | NaN | Inf | 7.13 | Inf | Inf | 5.40 | Inf | Inf | Inf | Inf | Inf | 6.17 |
| +QuaRot | 6.27 | 6.20 | 5.66 | 5.51 | 5.47 | 5.04 | 8.20 | 8.02 | 6.62 | 5.75 | 5.71 | 5.37 |
| +SpinQuant | 5.94 | 5.91 | 5.65 | 5.25 | 5.21 | 5.04 | 7.34 | 7.25 | 6.63 | 5.62 | 5.57 | 5.37 |
| +DuaRot | 5.79 | 5.74 | 5.65 | 5.13 | 5.12 | 5.03 | 7.22 | 7.13 | 6.62 | 5.55 | 5.50 | 5.37 |

Table 2: Average zero-shot accuracy (↑) of LLaMA and Mistral with RTN and GPTQ on PIQA, WinoGrande, HellaSwag, ARC-challenge, ARC-easy, OBQA, BoolQ and SIQA. Full results and more quantization results, including W4A4KV8, W4A8KV8, W4A8KV16 are in the Appendix.

| Method | LLaMA2-7B | | | LLaMA2-13B | | | LLaMA3-8B | | | Mistral-7B | | |
|---|---|---|---|---|---|---|---|---|---|---|---|---|
| Baseline | 64.06 | | | 66.41 | | | 67.17 | | | 68.33 | | |
| | 4-4-4 | 4-4-16 | 4-8-4 | 4-4-4 | 4-4-16 | 4-8-4 | 4-4-4 | 4-4-16 | 4-8-4 | 4-4-4 | 4-4-16 | 4-8-4 |
| RTN | 35.47 | 34.77 | 56.72 | 34.84 | 35.10 | 62.97 | 36.15 | 35.79 | 47.62 | 35.06 | 35.56 | 62.54 |
| +QuaRot | 53.89 | 53.95 | 59.09 | 60.58 | 60.93 | 63.79 | 55.87 | 56.55 | 63.09 | 63.32 | 63.58 | 66.16 |
| +SpinQuant | 58.96 | 58.21 | 61.08 | 62.92 | 63.98 | 65.72 | 61.41 | 62.30 | 64.28 | 65.36 | 65.48 | 67.08 |
| +DuaRot | 59.79 | 59.52 | 61.32 | 62.94 | 63.09 | 64.84 | 63.51 | 63.57 | 65.76 | 65.53 | 66.11 | 67.28 |
| GPTQ | 35.63 | 35.88 | 59.68 | 34.77 | 33.97 | 63.53 | 36.13 | 35.71 | 39.95 | 36.44 | 36.14 | 64.79 |
| +QuaRot | 59.96 | 60.46 | 62.94 | 63.49 | 63.53 | 65.38 | 60.46 | 61.05 | 65.96 | 65.11 | 65.03 | 67.35 |
| +SpinQuant | 60.67 | 60.11 | 63.14 | 64.41 | 64.57 | 65.54 | 63.46 | 63.68 | 66.19 | 65.99 | 66.39 | 67.46 |
| +DuaRot | 61.15 | 61.20 | 62.79 | 64.63 | 64.35 | 66.09 | 64.15 | 64.21 | 65.43 | 66.01 | 66.81 | 67.80 |

Following SpinQuant Liu et al. (2024), we also utilize 800 samples from WikiText-2 to optimize rotation matrices for 100 iterations and decay learning rate from 1.5 to 0 via cosine scheduler. We apply per-channel symmetric quantization to weight and set quantization ranges via a linear search to minimize the mean-squared error between quantized and full-precision weights. The activation and key-value cache (KV Cache) are applied with asymmetric min-max dynamic quantization with per-token activation quantization and group size 128, which is the same to $D_{head}$ for the key-value quantization. We use 128 samples from the WikiText-2 (Merity et al., 2016) training set as the calibration dataset for GPTQ quantization and the sequence length is set to 2048. In the clipping settings, we set the activation clip ratio and KV Cache clipping ratio to (0.75, 0.95) for RTN quantization and (0.98, 0.96) for GPTQ quantization respectively.

## 4.2 ACCURACY RESULTS

**Language Generation Tasks.** We first evaluate the accuracy of DuaRot on the language generation task. We conduct experiments in challenging W4A4KV4 and W4A4KV16 and popular W4A8KV4 quantization settings. Table 1 shows the PPL for WikiText-2 on the LLaMA and the Mistral-7B models. We compare DuaRot with rotation-based methods, including QuaRot and SpinQuant and quantize weights through RTN and GPTQ, respectively. As seen, benefiting from more effective outlier mitigation compared to SpinQuant, DuaRot achieved consistent improvements across different models and various quantization configurations, especially for 4-bit activation quantization. In the most challenging W4A4KV4 quantization, DuaRot achieved significant improvements over SpinQuant. For example, on the LLaMA3-8B model, which is well-known for its quantization challenges, DuaRot achieved 0.51 and 0.12 PPL improvement over SpinQuant when using RTN

Table 3: Ablation studies for hardware-aware matrix configuration strategy and dual rotation. All models are quantized using RTN. We reported PPL results for WikiText-2. Dual: Dual Rotation.

| Dual | Hardware | LLaMA2-7B | | | LLaMA2-13B | | | LLaMA3-8B | | | Mistral-7B | | |
|---|---|---|---|---|---|---|---|---|---|---|---|---|---|
| | | 4-4-4 | 4-4-16 | 4-8-4 | 4-4-4 | 4-4-16 | 4-8-4 | 4-4-4 | 4-4-16 | 4-8-4 | 4-4-4 | 4-4-16 | 4-8-4 |
| ✗ | ✗ | 6.20 | 6.17 | 5.56 | 5.51 | 5.40 | 4.97 | 8.00 | 7.82 | 6.79 | 5.60 | 5.58 | 5.28 |
| ✓ | ✗ | 6.01 | 5.99 | 5.54 | 5.40 | 5.30 | 4.96 | 7.70 | 7.60 | 6.78 | 5.54 | 5.44 | 5.26 |
| ✗ | ✓ | 5.92 | 5.87 | 5.54 | 5.30 | 5.22 | 4.96 | 7.56 | 7.44 | 6.79 | 5.50 | 5.47 | 5.28 |
| ✓ | ✓ | 5.88 | 5.86 | 5.51 | 5.27 | 5.19 | 4.94 | 7.49 | 7.41 | 6.76 | 5.48 | 5.43 | 5.24 |

and GPTQ. Meanwhile, from Table 1, we can find that both training rotational matrices and GPTQ can significantly enhances the performance of Rotated LLM in 4-bit activation quantization, with all models achieving SOTA PPL performance by combining both. In contrast, for 8-bit activation quantization with GPTQ, the activation quantization is relatively easy to quantize, GPTQ+QuaRot has achieved superior performance. Using trained rotation matrices to further eliminate outliers does not produce noticeable effects.

**Zero-shot Common Sense Reasoning Tasks.** Next, we evaluate DuaRot on eight zero-shot common sense reasoning tasks. As shown in Table 2, DuaRot also achieves comparable average score on the above tasks. For example, on the LLaMA3-8B model, compared to SpinQuant, DuaRot achieved accuracy improvements of 2.10, 1.27, and 1.48 in the W4A4KV4, W4A4KV16, and W4A8KV4 RTN quantization, respectively. However, we also noticed that further mitigation of outliers might lead to a degradation in the model's zero-shot capabilities. For instance, on the LLaMA2-13B model, although DuaRot achieves PPL improvements of 0.21 and 0.09 on WikiText-2 in the W4A4KV16 RTN and GPTQ quantization compared to SpinQuant respectively, the zero-shot accuracy actually decreased by 0.89 and 0.22. This could be due to the further suppression of outliers on the target dataset, causing the rotation matrix to overfit the data under the quantization configuration. How to optimize the rotation matrices while retaining or enhancing the model's zero-shot capabilities is an interesting topic and will be our future direction.

**Compare with Scaling-based Methods.** It is worth mentioning that in this paper, we do not compare our method with scaling-based approaches such as SmoothQuant (Xiao et al., 2023) and OmniQuant (Shao et al., 2023). This is because we find that when the activation values are quantized to 4 bits, including W4A4KV4 and W4A4KV16, scaling-based methods almost invariably fail to perform, while rotation-based methods can still perform effectively. For example, OmniQuant achieved a PPL of 14.3 on WikiText2 with the LLaMA2-7B model under the W4A4KV16 configuration, which lagged significantly behind QuaRot, achieving 8.69 with RTN and 6.20 with GPTQ.

## 4.3 ABLATION STUDIES

**Hardware-aware matrix configuration strategy and dual rotation.** In Table 3, we conduct extensive experiments on RTN quantization and WikiText-2 to investigate the effectiveness of different components in DuaRot, named hardware-aware matrix configuration strategy and dual rotation. The baseline is reproduced using SpinQuant publicly released codebase.

With our training strategies, we find that extending $R_3$ and $R_4$ into the trainable space can further suppress outliers compared to Hadamard matrix and enhance the model's performance. For example, under the W4A4KV4 and W4A4KV16 quantization configuration, extending $R_4$ into the trainable space can achieve a major improvement 0.44 and 0.38 respectively, in PPL on the LLaMA3-8B model. Furthermore, in the W4A4KV4 configuration, setting $R_3$ as trainable parameters can further reduce the quantization loss of Query and Key, improving the quantization performance of the attention. It is worth noting that although extending $R_3$ and $R_4$ into the trainable space prevents the use of WHT for acceleration, the powerful computational capabilities of GPUs enable efficient Matmul and the model's speed will not decrease. In fact, this approach can help improve the model's speed during the decoding phase as shown in Figure 4(a).

Dual Rotation also provides gains in improving the model's performance. During our experiments, we find that the ability of $R_1$ to suppress outliers is also crucial for enhancing the quantization performance of Rotated LLMs. By applying dual rotation, we can achieve better outlier suppression

through both global and local rotations. further improving the model's quantization performance. It can be observed combining DuaRot and hardware-aware matrix configuration achieves the best results among the all models and settings.

**Reparameterization matrix size.** We conduct ablation studies involving local rotation matrix size $d$ on LLaMA3-8B with W4A4KV4 quantization. We select six different settings of $d$, which vary from 32 to 1024 and present the PPL results in Figure 5. A intuition is that a smaller $d$ focuses on eliminating outliers within fine-grained, but this might lead to outliers being concentrated within the group, failed to be dispersed to other groups through rotation. On the other hand, although a larger $d$ can disperse outliers to a greater extent, dual rotations might introduce instability during the training, potentially leading to a decline in the quantized model's performance after rotation. As seen, LLaMA3-8B achieves best results, 7.49 PPL on WikiText-2 with $d = 64$. Both $d = 1024$ and $d = 32$ result in a decline in the model's performance compared to $d = 64$. Based on this, we set $d$ to 64 by default in this paper.

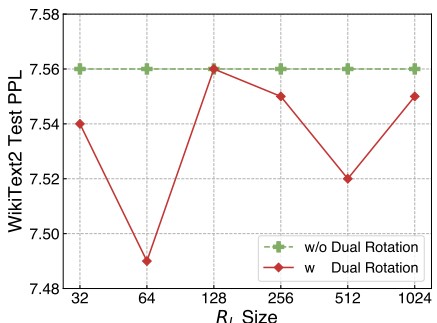

Figure 5: Ablation study on local rotation matrix size for W4A4KV4 LLaMA3-8B with RTN.

Although this is not optimal, our method still achieves a considerable improvement in accuracy across various LLM models and quantization settings and methods.

Table 4: PPL results of LLaMA and Misral models with RTN and W4A4KV4 on WikiText-2 and C4 dataset. For each model, we conducted training on the WikiText-2 and C4 datasets respectively.

| Model | LLaMA2-7B | | | | LLaMA2-13B | | | | LLaMA3-8B | | | | Mistral-7B | | | |
|---|---|---|---|---|---|---|---|---|---|---|---|---|---|---|---|---|
| Train | Wiki | | C4 | | Wiki | | C4 | | Wiki | | C4 | | Wiki | | C4 | |
| Test | Wiki | C4 | Wiki | C4 | Wiki | C4 | Wiki | C4 | Wiki | C4 | Wiki | C4 | Wiki | C4 | Wiki | C4 |
| PPL | 5.88 | 8.17 | 6.36 | 7.99 | 5.27 | 7.33 | 5.60 | 7.21 | 7.49 | 11.42 | 7.92 | 11.19 | 5.48 | 8.50 | 5.77 | 8.35 |

Table 5: Zero-shot accuracy of LLaMA and Misral models with RTN and W4A4KV4 on PIQA, WinoGrande (WG), HellaSwag (HS), ARC-challenge (ARC-c), ARC-easy (ARC-e), OBQA, BoolQ and SIQA. For each model, we conducted training on the WikiText-2 and C4 datasets respectively.

| Model | Train | PIQA | WG | HS | ARC-c | ARC-e | OBQA | BoolQ | SIQA | Avg. |
|---|---|---|---|---|---|---|---|---|---|---|
| LLaMA2-7B | WikiText-2 | 74.97 | 65.67 | 71.43 | 41.30 | 69.87 | 39.60 | 72.54 | 42.94 | 59.79 |
| | C4 | 76.28 | 65.19 | 71.82 | 39.76 | 66.41 | 40.60 | 70.76 | 42.43 | 59.16 |
| LLaMA2-13B | WikiText-2 | 78.13 | 69.06 | 76.55 | 46.33 | 73.99 | 41.80 | 72.08 | 45.60 | 62.94 |
| | C4 | 78.35 | 69.85 | 76.17 | 45.22 | 73.74 | 44.00 | 78.38 | 45.55 | 63.91 |
| LLaMA3-8B | WikiText-2 | 78.40 | 66.54 | 76.36 | 48.55 | 74.45 | 43.00 | 75.38 | 45.39 | 63.51 |
| | C4 | 78.94 | 68.19 | 74.87 | 46.08 | 71.89 | 42.00 | 76.94 | 44.88 | 62.97 |
| Mistral-7B | WikiText-2 | 81.56 | 70.48 | 78.56 | 50.51 | 78.07 | 41.00 | 80.76 | 43.30 | 65.53 |
| | C4 | 80.03 | 72.30 | 78.64 | 49.40 | 76.52 | 44.80 | 82.60 | 45.24 | 66.19 |

**Discussion to the calibration set.** We further discuss the impact of the calibration dataset distribution on model performance. We select C4 dataset and alternately use C4 and WikiText-2 as the training and test sets to evaluate corresponding PPL. As seen in Table 4, DuaRot performance on PPL depends on specific calibration set. Training and testing on different calibration datasets can lead to a significant decrease in the quantized model's PPL on the target dataset. For example, for LLaMA3-8B with W4A4KV4 quantization, the PPL when trained and tested on WikiText-2 improves by 0.43 compared to when trained on C4. We believe this is because, although training the rotation matrix keeps rotational invariance, *e.g.* full-precision output will not change, the rotation matrix will fit the quantization distribution of the specific calibration data to achieve better performance on the target dataset.

Additionally, we compare the performance on zero-shot common sense reasoning tasks. As seen in Table 5, the impact of the training dataset on average zero-shot performance is much smaller than the PPL on the target dataset. From this perspective, the reasonableness of evaluating the quantized model only based on PPL is questionable. Meanwhile, there are still some interesting phenomena for some specific models and data. For Arc-challenge, all models trained on C4 have slightly lower accuracy than WikiText-2. For BoolQ, all models except LLaMA2-7B achieve better results when trained based on C4, and in particular LLaMA2-13B improves the accuracy by 6.2%. Considering the importance that the dataset quality will have on the effectiveness of the model during Supervised fine-tuning (SFT) (Zhou et al., 2024), we believe that choosing the appropriate calibration dataset for a specific scenario is equally crucial to the performance of the quantized model. How to further improve the generalization capability of the trained model from the perspective of calibration dataset is a direction worth exploring in the future.

## 5 CONCLUSION

In this paper, we propose a *Dual Rotation* method, DuaRot, to achieve more advanced elimination for activation outliers while retaining the efficiency of quantized models. DuaRot follows a reparameterization method for rotational matrices. During the training process, both global and local rotational matrices are trained separately, with the former refine broader activation distributions and the latter focusing on finer-grained details. During inference, these matrices can be merged without introducing any computational overhead, while still maintaining the rotational invariance. Moreover, DuaRot employs a hardware-aware matrix configuration strategy. By comparing the runtime performance of WHT and Matmul across different matrix sizes, DuaRot extends the matrices requiring online computation into a trainable space. This approach achieves better accuracy performance without sacrificing inference speed. Extensive experiments have demonstrated the effectiveness of DuaRot across various models, quantization configurations, and weight quantization techniques. For example, DuaRot achieves 7.49 and 7.41 PPL to LLaMA3-8B under W4A4KV4 RTN quantizations, improving by 0.51 and 0.41 over the state-of-the-art respectively.

## 6 LIMITATIONS

In this study, we introduce a dual rotation and hardware-aware matrix configuration strategy. This method achieved significant improvements in the target dataset by fine-tuning the rotation matrix on the calibration set. However, this method still has many issues expect calibration dataset as mentioned in Section 4.3:

**Training Cost.** Since each MHA and FFN is affected by $R_1$, we can't optimize $R_1$ using a layer-by-layer or block-by-block approach. Optimizing $R_1$ directly via gradient is a straightforward method. Although the optimizer needn't to store weight state and the memory overhead required for training is significantly reduced compared to QAT methods, both SpinQuant and DuaRot still require loading the entire model into the GPU and updating $R_1$ using gradient methods. Compared to GPTQ, training-based methods are still expensive. For example, training LLaMA2-70B requires at least four NVIDIA A100 80GB GPUs. In the future, reducing the training cost remains a direction worth exploring. For instance, calibrating $R_1$ in a cheap and efficient way, and then using a block-by-block approach to train $R_2$, $R_3$, and $R_4$. This approach can bring the cost of LLaMA2-70B down to the same level as GPTQ.

**Optimize Efficiency.** Although Cayley optimization can retain the orthogonality of the matrix during training, it is achieved through a fixed-point iteration method. This iterative approach inevitably increases the training overhead. Additionally, during training, we find that the rotational matrix $R_1$ after training is only optimized around the randomized Hadamard matrix, which denotes that the gradients obtained using Cayley optimization are essentially close to the identity matrix $I$. It will make the rotational matrix highly dependent on initialization and cannot optimize efficiently over the entire orthogonal group $SO(n)$, reducing the optimization efficiency of the model. In the future, studying how to learn the rotational matrix more efficiently over the entire orthogonal group will further help reduce training overhead.

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

# A  FULL RESULTS

Table 6: Comparison of the WikiText-2 perplexity (↓) results for LLaMA and Mistral. The 4-4-4, 4-4-8, 4-4-16, 4-8-4, 4-8-8, 4-8-16 represent W4A4KV4, W4A4KV8, W4A4KV16, W4A4KV4, W4A4KV8 and W4A4KV16, respectively. We show the failed GPTQ experiments using NaN and the perplexity results>100 by Inf. Results for RTN, GPTQ and QuaRot (Ashkboos et al., 2024b) are obtained using QuaRot publicly released codebase. Results for SpinQuant (Liu et al., 2024) are obtained using their publicly released codebase.

| Method | LLaMA2-7B | | | LLaMA2-13B | | | LLaMA3-8B | | | Mistral-7B | | |
|---|---|---|---|---|---|---|---|---|---|---|---|---|
| Baseline | 5.47 | | | 4.88 | | | 6.13 | | | 5.25 | | |
| | 4-4-4 | 4-4-8 | 4-4-16 | 4-4-4 | 4-4-8 | 4-4-16 | 4-4-4 | 4-4-8 | 4-4-16 | 4-4-4 | 4-4-8 | 4-4-16 |
| RTN | NaN | NaN | NaN | Inf | Inf | Inf | Inf | Inf | Inf | Inf | Inf | Inf |
| +QuaRot | 9.04 | 8.70 | 8.69 | 6.31 | 6.22 | 6.23 | 11.06 | 10.5 | 10.47 | 6.26 | 6.19 | 6.19 |
| +SpinQuant | 6.20 | 6.15 | 6.17 | 5.51 | 5.47 | 5.40 | 8.00 | 7.80 | 7.82 | 5.60 | 5.58 | 5.58 |
| **+DuaRot** | 5.88 | 5.85 | 5.86 | 5.27 | 5.25 | 5.19 | 7.49 | 7.40 | 7.41 | 5.48 | 5.42 | 5.43 |
| GPTQ | NaN | NaN | Inf | Inf | Inf | Inf | Inf | Inf | Inf | Inf | Inf | Inf |
| +QuaRot | 6.27 | 6.19 | 6.20 | 5.51 | 5.46 | 5.47 | 8.20 | 8.03 | 8.02 | 5.75 | 5.71 | 5.71 |
| +SpinQuant | 5.94 | 5.88 | 5.91 | 5.25 | 5.21 | 5.21 | 7.34 | 7.24 | 7.25 | 5.62 | 5.58 | 5.57 |
| **+DuaRot** | 5.79 | 5.75 | 5.74 | 5.13 | 5.11 | 5.12 | 7.22 | 7.17 | 7.13 | 5.55 | 5.50 | 5.50 |
| | 4-8-4 | 4-8-8 | 4-8-16 | 4-8-4 | 4-8-8 | 4-8-16 | 4-8-4 | 4-8-8 | 4-8-16 | 4-8-4 | 4-8-8 | 4-8-16 |
| RTN | 7.92 | 7.37 | 7.37 | 5.79 | 5.47 | 5.47 | 19.38 | 10.89 | 10.88 | 12.70 | 11.21 | 10.65 |
| +QuaRot | 6.89 | 6.75 | 6.75 | 5.51 | 5.46 | 5.46 | 7.81 | 7.64 | 7.64 | 5.70 | 5.65 | 5.65 |
| +SpinQuant | 5.56 | 5.51 | 5.52 | 4.97 | 4.94 | 4.93 | 6.79 | 6.71 | 6.70 | 5.28 | 5.24 | 5.25 |
| **+DuaRot** | 5.51 | 5.49 | 5.48 | 4.94 | 4.91 | 4.91 | 6.76 | 6.67 | 6.67 | 5.24 | 5.22 | 5.24 |
| GPTQ | 7.13 | 6.55 | 6.60 | 5.40 | 5.21 | 5.21 | Inf | Inf | Inf | 6.17 | 6.01 | 6.01 |
| +QuaRot | 5.66 | 5.61 | 5.61 | 5.04 | 5.00 | 5.01 | 6.62 | 6.52 | 6.51 | 5.37 | 5.34 | 5.34 |
| +SpinQuant | 5.65 | 5.61 | 5.62 | 5.04 | 5.00 | 5.00 | 6.63 | 6.54 | 6.55 | 5.37 | 5.34 | 5.34 |
| **+DuaRot** | 5.65 | 5.61 | 5.61 | 5.03 | 5.00 | 5.00 | 6.62 | 6.53 | 6.51 | 5.37 | 5.34 | 5.34 |

Table 7: Zero-shot accuracy of LLaMA2-7B with RTN and GPTQ on PIQA, WinoGrande (WG), HellaSwag (HS), ARC-challenge (ARC-c), ARC-easy (ARC-e), OBQA, BoolQ and SIQA.

| W-A-KV | Method | PIQA | WG | HS | ARC-c | ARC-e | OBQA | BoolQ | SIQA | Avg. |
|---|---|---|---|---|---|---|---|---|---|---|
| 16-16-16 | Baseline | 79.11 | 69.22 | 76.00 | 46.33 | 74.54 | 44.20 | 76.97 | 46.11 | 64.06 |
| 4-4-4 | RTN | 50.60 | 51.14 | 26.08 | 26.96 | 27.02 | 25.80 | 42.20 | 33.93 | 35.47 |
| | +QuaRot | 70.29 | 60.77 | 63.87 | 36.86 | 59.55 | 33.40 | 66.33 | 40.02 | 53.89 |
| | +SpinQuant | 75.52 | 63.85 | 70.91 | 39.76 | 65.07 | 40.60 | 72.08 | 43.91 | 58.96 |
| | **+DuaRot** | 74.97 | 65.67 | 71.43 | 41.30 | 69.87 | 39.60 | 72.54 | 42.94 | 59.79 |
| 4-4-8 | RTN | 50.65 | 50.51 | 25.84 | 24.66 | 26.81 | 25.00 | 42.11 | 34.08 | 34.96 |
| | +QuaRot | 69.97 | 61.09 | 65.30 | 35.67 | 60.61 | 35.60 | 69.72 | 40.58 | 54.82 |
| | +SpinQuant | 75.79 | 63.30 | 70.92 | 38.57 | 61.74 | 39.40 | 71.68 | 41.66 | 57.88 |
| | **+DuaRot** | 76.50 | 65.27 | 72.36 | 41.47 | 66.84 | 39.80 | 71.04 | 42.63 | 59.49 |
| 4-4-16 | RTN | 50.22 | 48.46 | 25.79 | 24.74 | 25.67 | 27.40 | 41.87 | 34.03 | 34.77 |
| | +QuaRot | 70.02 | 60.06 | 64.80 | 35.92 | 59.22 | 33.00 | 68.50 | 40.07 | 53.95 |
| | +SpinQuant | 75.84 | 63.06 | 70.90 | 39.85 | 64.39 | 39.60 | 70.98 | 41.04 | 58.21 |
| | **+DuaRot** | 75.79 | 65.35 | 71.55 | 41.38 | 66.79 | 37.60 | 74.34 | 43.35 | 59.52 |
| 4-8-4 | RTN | 74.59 | 60.93 | 67.87 | 39.33 | 66.67 | 35.60 | 66.21 | 42.58 | 56.72 |
| | +QuaRot | 76.01 | 64.96 | 70.77 | 40.70 | 67.38 | 37.20 | 73.33 | 42.37 | 59.09 |
| | +SpinQuant | 77.31 | 65.98 | 74.25 | 42.32 | 70.45 | 42.20 | 73.61 | 42.53 | 61.08 |
| | **+DuaRot** | 76.71 | 66.54 | 74.29 | 42.83 | 70.83 | 42.60 | 73.09 | 43.65 | 61.32 |
| 4-8-8 | RTN | 75.57 | 65.27 | 70.72 | 43.09 | 67.93 | 40.60 | 67.92 | 43.14 | 59.28 |
| | +QuaRot | 75.79 | 65.90 | 71.34 | 41.04 | 68.48 | 37.80 | 73.39 | 42.22 | 59.50 |
| | +SpinQuant | 77.53 | 66.61 | 74.11 | 42.66 | 70.08 | 42.60 | 74.13 | 42.53 | 61.28 |
| | **+DuaRot** | 77.91 | 66.77 | 74.05 | 44.71 | 72.35 | 43.00 | 74.10 | 42.17 | 61.88 |
| 4-8-16 | RTN | 75.95 | 64.72 | 70.58 | 42.75 | 68.31 | 42.00 | 67.52 | 42.68 | 59.31 |
| | +QuaRot | 76.33 | 65.51 | 71.34 | 40.70 | 68.39 | 37.60 | 73.46 | 42.32 | 59.46 |
| | +SpinQuant | 77.58 | 67.25 | 74.44 | 43.00 | 69.49 | 41.60 | 74.89 | 42.32 | 61.32 |
| | **+DuaRot** | 77.86 | 67.01 | 74.75 | 45.22 | 72.22 | 42.80 | 74.71 | 44.58 | 62.39 |
| 4-4-4 | GPTQ | 49.08 | 48.30 | 25.77 | 25.77 | 27.95 | 25.40 | 49.48 | 33.32 | 35.63 |
| | +QuaRot | 77.15 | 65.82 | 72.76 | 41.47 | 69.44 | 37.80 | 71.87 | 43.35 | 59.96 |
| | +SpinQuant | 76.66 | 65.98 | 72.78 | 42.06 | 70.92 | 38.40 | 73.82 | 44.73 | 60.67 |
| | **+DuaRot** | 76.50 | 67.09 | 72.69 | 42.15 | 71.51 | 42.00 | 72.60 | 44.68 | 61.15 |
| 4-4-8 | GPTQ | 50.71 | 49.25 | 26.75 | 26.54 | 27.57 | 28.00 | 47.22 | 33.88 | 36.24 |
| | +QuaRot | 76.39 | 65.67 | 72.88 | 41.81 | 69.61 | 39.60 | 73.18 | 43.55 | 60.34 |
| | +SpinQuant | 77.09 | 67.48 | 73.39 | 43.00 | 69.87 | 40.40 | 75.20 | 43.81 | 61.28 |
| | **+DuaRot** | 76.22 | 65.11 | 72.90 | 43.60 | 71.09 | 41.80 | 75.08 | 44.47 | 61.28 |
| 4-4-16 | GPTQ | 50.27 | 48.15 | 26.26 | 26.96 | 27.23 | 25.60 | 47.68 | 34.90 | 35.88 |
| | +QuaRot | 77.31 | 65.43 | 72.95 | 41.55 | 70.03 | 39.00 | 73.46 | 43.91 | 60.46 |
| | +SpinQuant | 75.24 | 66.14 | 72.82 | 40.44 | 68.77 | 39.20 | 74.53 | 43.76 | 60.11 |
| | **+DuaRot** | 76.82 | 66.30 | 72.77 | 42.66 | 72.39 | 41.20 | 72.17 | 45.29 | 61.20 |
| 4-8-4 | GPTQ | 76.82 | 63.61 | 71.54 | 40.53 | 68.14 | 43.00 | 70.80 | 42.99 | 59.68 |
| | +QuaRot | 78.56 | 68.67 | 75.08 | 43.77 | 73.11 | 43.20 | 75.87 | 45.29 | 62.94 |
| | +SpinQuant | 78.07 | 69.22 | 74.66 | 45.65 | 73.57 | 43.80 | 75.02 | 45.14 | 63.14 |
| | **+DuaRot** | 78.78 | 68.51 | 74.69 | 43.09 | 72.77 | 43.20 | 76.02 | 45.29 | 62.79 |
| 4-8-8 | GPTQ | 77.91 | 66.06 | 73.40 | 42.15 | 70.66 | 42.20 | 72.51 | 43.71 | 61.08 |
| | +QuaRot | 78.94 | 69.46 | 75.09 | 43.52 | 73.53 | 43.00 | 76.21 | 45.14 | 63.11 |
| | +SpinQuant | 77.31 | 67.32 | 74.92 | 42.92 | 72.35 | 43.00 | 74.62 | 44.83 | 62.16 |
| | **+DuaRot** | 78.18 | 68.75 | 74.50 | 44.11 | 73.23 | 43.00 | 76.64 | 45.80 | 63.03 |
| 4-8-16 | GPTQ | 77.20 | 66.85 | 73.33 | 42.32 | 69.95 | 42.40 | 73.43 | 43.76 | 61.15 |
| | +QuaRot | 78.84 | 68.98 | 75.12 | 43.94 | 73.53 | 42.80 | 76.18 | 45.19 | 63.07 |
| | +SpinQuant | 77.86 | 68.03 | 74.85 | 43.43 | 72.81 | 43.20 | 74.16 | 44.78 | 62.39 |
| | **+DuaRot** | 77.97 | 68.51 | 74.68 | 43.94 | 73.15 | 43.20 | 74.22 | 45.04 | 62.59 |

Table 8: Zero-shot accuracy of LLaMA2-13B with RTN and GPTQ on PIQA, WinoGrande (WG), HellaSwag (HS), ARC-challenge (ARC-c), ARC-easy (ARC-e), OBQA, BoolQ and SIQA.

| W-A-KV | Method | PIQA | WG | HS | ARC-c | ARC-e | OBQA | BoolQ | SIQA | Avg. |
|---|---|---|---|---|---|---|---|---|---|---|
| 16-16-16 | Baseline | 80.52 | 72.22 | 79.38 | 48.98 | 77.53 | 45.20 | 80.09 | 47.34 | 66.41 |
| 4-4-4 | RTN | 47.99 | 50.36 | 26.55 | 27.99 | 26.35 | 25.00 | 39.33 | 35.11 | 34.84 |
| | +QuaRot | 76.66 | 66.30 | 72.19 | 43.77 | 70.08 | 39.60 | 73.70 | 42.32 | 60.58 |
| | +SpinQuant | 77.97 | 65.82 | 75.39 | 45.82 | 74.41 | 42.80 | 75.78 | 45.39 | 62.92 |
| | **+DuaRot** | 78.13 | 69.06 | 76.55 | 46.33 | 73.99 | 41.80 | 72.08 | 45.60 | 62.94 |
| 4-4-8 | RTN | 49.13 | 51.22 | 26.35 | 25.68 | 26.39 | 25.20 | 38.35 | 35.36 | 34.71 |
| | +QuaRot | 77.75 | 65.67 | 72.71 | 44.37 | 71.25 | 39.60 | 74.77 | 41.45 | 60.95 |
| | +SpinQuant | 78.18 | 67.32 | 75.41 | 46.08 | 74.33 | 42.80 | 77.77 | 47.49 | 63.67 |
| | **+DuaRot** | 78.56 | 68.90 | 76.82 | 45.82 | 72.85 | 43.80 | 78.01 | 45.60 | 63.80 |
| 4-4-16 | RTN | 50.92 | 49.88 | 26.01 | 29.01 | 26.05 | 25.40 | 38.90 | 34.60 | 35.10 |
| | +QuaRot | 76.55 | 66.22 | 72.86 | 44.37 | 70.33 | 40.60 | 74.62 | 41.91 | 60.93 |
| | +SpinQuant | 78.13 | 69.14 | 76.08 | 46.33 | 75.08 | 43.40 | 77.65 | 46.06 | 63.98 |
| | **+DuaRot** | 78.35 | 67.72 | 76.52 | 45.56 | 73.53 | 43.20 | 75.14 | 44.68 | 63.09 |
| 4-8-4 | RTN | 77.86 | 67.01 | 75.64 | 45.90 | 74.20 | 40.80 | 76.64 | 45.75 | 62.97 |
| | +QuaRot | 79.11 | 69.53 | 75.84 | 46.50 | 74.28 | 42.20 | 78.35 | 44.47 | 63.79 |
| | +SpinQuant | 79.98 | 70.88 | 77.95 | 48.72 | 76.77 | 45.40 | 79.30 | 46.78 | 65.72 |
| | **+DuaRot** | 79.27 | 70.96 | 78.50 | 47.01 | 75.08 | 45.20 | 78.04 | 44.63 | 64.84 |
| 4-8-8 | RTN | 78.67 | 71.35 | 77.24 | 48.55 | 75.76 | 43.60 | 79.20 | 46.72 | 65.14 |
| | +QuaRot | 78.89 | 68.75 | 75.99 | 47.27 | 74.92 | 43.40 | 79.36 | 44.63 | 64.15 |
| | +SpinQuant | 80.20 | 70.80 | 78.37 | 49.57 | 77.02 | 44.80 | 79.11 | 46.32 | 65.77 |
| | **+DuaRot** | 80.03 | 71.03 | 78.71 | 47.78 | 76.52 | 44.40 | 77.49 | 44.78 | 65.09 |
| 4-8-16 | RTN | 79.00 | 71.03 | 77.37 | 48.21 | 75.29 | 43.60 | 79.05 | 46.62 | 65.02 |
| | +QuaRot | 78.89 | 68.90 | 76.00 | 46.84 | 74.92 | 43.40 | 78.87 | 44.68 | 64.06 |
| | +SpinQuant | 80.03 | 69.06 | 78.45 | 49.32 | 77.23 | 44.60 | 79.05 | 46.52 | 65.53 |
| | **+DuaRot** | 79.11 | 69.61 | 78.93 | 47.61 | 76.05 | 43.00 | 78.26 | 45.04 | 64.70 |
| 4-4-4 | GPTQ | 49.95 | 49.72 | 25.86 | 27.39 | 26.81 | 23.40 | 40.06 | 34.95 | 34.77 |
| | +QuaRot | 77.75 | 69.93 | 75.82 | 45.65 | 73.65 | 43.20 | 76.73 | 45.19 | 63.49 |
| | +SpinQuant | 78.78 | 70.24 | 76.63 | 47.35 | 75.93 | 44.40 | 75.81 | 46.11 | 64.41 |
| | **+DuaRot** | 79.22 | 70.96 | 77.35 | 46.08 | 76.01 | 43.40 | 78.29 | 45.70 | 64.63 |
| 4-4-8 | GPTQ | 48.37 | 51.30 | 25.85 | 26.79 | 26.85 | 23.60 | 39.85 | 34.54 | 34.64 |
| | +QuaRot | 77.53 | 69.61 | 76.13 | 46.59 | 74.62 | 44.60 | 76.91 | 45.60 | 63.95 |
| | +SpinQuant | 78.73 | 69.69 | 77.48 | 48.04 | 76.52 | 45.60 | 77.61 | 45.96 | 64.95 |
| | **+DuaRot** | 79.43 | 69.77 | 77.40 | 47.01 | 75.13 | 42.60 | 77.95 | 45.75 | 64.38 |
| 4-4-16 | GPTQ | 47.39 | 47.99 | 26.08 | 25.68 | 26.94 | 24.40 | 39.39 | 33.88 | 33.97 |
| | +QuaRot | 78.51 | 69.77 | 75.55 | 45.65 | 74.49 | 42.00 | 77.09 | 45.19 | 63.53 |
| | +SpinQuant | 78.45 | 68.59 | 77.26 | 47.44 | 75.38 | 44.40 | 77.77 | 47.29 | 64.57 |
| | **+DuaRot** | 80.03 | 68.27 | 76.67 | 47.70 | 75.08 | 42.20 | 78.01 | 46.83 | 64.35 |
| 4-8-4 | GPTQ | 77.09 | 70.32 | 75.63 | 45.82 | 73.65 | 42.60 | 77.09 | 46.06 | 63.53 |
| | +QuaRot | 79.11 | 71.74 | 78.34 | 46.76 | 76.77 | 44.80 | 79.20 | 46.32 | 65.38 |
| | +SpinQuant | 80.09 | 70.96 | 78.15 | 47.35 | 76.94 | 44.60 | 79.60 | 46.62 | 65.54 |
| | **+DuaRot** | 79.76 | 71.51 | 78.14 | 50.17 | 77.65 | 45.00 | 79.51 | 46.98 | 66.09 |
| 4-8-8 | GPTQ | 79.11 | 70.24 | 76.71 | 46.33 | 75.38 | 43.80 | 76.97 | 45.70 | 64.28 |
| | +QuaRot | 79.82 | 71.43 | 78.42 | 47.87 | 76.81 | 44.40 | 79.17 | 46.72 | 65.58 |
| | +SpinQuant | 80.03 | 71.35 | 78.33 | 48.46 | 76.39 | 43.40 | 79.08 | 46.37 | 65.43 |
| | **+DuaRot** | 79.71 | 73.01 | 78.28 | 49.74 | 77.44 | 46.00 | 79.60 | 46.93 | 66.34 |
| 4-8-16 | GPTQ | 79.16 | 70.48 | 76.60 | 46.93 | 75.63 | 43.20 | 76.67 | 45.75 | 64.30 |
| | +QuaRot | 79.87 | 71.35 | 78.44 | 47.95 | 76.81 | 44.20 | 79.20 | 46.52 | 65.54 |
| | +SpinQuant | 80.36 | 71.51 | 78.51 | 49.15 | 77.40 | 44.20 | 79.54 | 46.72 | 65.92 |
| | **+DuaRot** | 79.65 | 70.96 | 78.49 | 49.23 | 76.26 | 44.20 | 78.69 | 47.34 | 65.60 |

Table 9: Zero-shot accuracy of LLaMA3-8B with RTN and GPTQ on PIQA, WinoGrande (WG), HellaSwag (HS), ARC-challenge (ARC-c), ARC-easy (ARC-e), OBQA, BoolQ and SIQA.

| W-A-KV | Method | PIQA | WG | HS | ARC-c | ARC-e | OBQA | BoolQ | SIQA | Avg. |
|---|---|---|---|---|---|---|---|---|---|---|
| 16-16-16 | Baseline | 80.79 | 72.53 | 79.15 | 53.41 | 77.74 | 45.00 | 81.62 | 47.08 | 67.17 |
| 4-4-4 | RTN | 50.71 | 48.86 | 27.06 | 24.66 | 28.45 | 27.60 | 47.22 | 34.60 | 36.15 |
| | +QuaRot | 70.24 | 63.69 | 68.70 | 38.05 | 60.65 | 35.20 | 69.14 | 41.30 | 55.87 |
| | +SpinQuant | 75.90 | 66.30 | 74.16 | 44.54 | 70.58 | 42.40 | 72.60 | 44.78 | 61.41 |
| | **+DuaRot** | 78.40 | 66.54 | 76.36 | 48.55 | 74.45 | 43.00 | 75.38 | 45.39 | 63.51 |
| 4-4-8 | RTN | 52.23 | 51.46 | 27.79 | 24.83 | 27.48 | 25.60 | 47.16 | 33.06 | 36.20 |
| | +QuaRot | 71.60 | 63.85 | 69.38 | 36.69 | 59.55 | 37.60 | 68.96 | 42.43 | 56.26 |
| | +SpinQuant | 77.04 | 68.98 | 75.00 | 44.03 | 70.12 | 39.60 | 75.11 | 43.96 | 61.73 |
| | **+DuaRot** | 78.84 | 70.09 | 76.70 | 48.98 | 77.06 | 42.40 | 77.46 | 46.57 | 64.76 |
| 4-4-16 | RTN | 51.90 | 48.78 | 28.06 | 22.70 | 28.79 | 23.80 | 48.20 | 34.08 | 35.79 |
| | +QuaRot | 72.96 | 63.22 | 69.36 | 37.20 | 60.86 | 35.40 | 71.10 | 42.32 | 56.55 |
| | +SpinQuant | 77.75 | 68.59 | 74.92 | 43.86 | 72.05 | 41.20 | 74.92 | 45.09 | 62.30 |
| | **+DuaRot** | 77.69 | 67.72 | 75.85 | 47.78 | 73.99 | 42.60 | 77.34 | 45.60 | 63.57 |
| 4-8-4 | RTN | 60.39 | 55.72 | 52.24 | 30.38 | 48.27 | 31.00 | 62.29 | 40.63 | 47.62 |
| | +QuaRot | 77.53 | 71.11 | 75.67 | 43.77 | 72.22 | 40.60 | 79.33 | 44.47 | 63.09 |
| | +SpinQuant | 78.78 | 73.01 | 77.79 | 49.83 | 72.52 | 41.60 | 75.84 | 44.83 | 64.28 |
| | **+DuaRot** | 79.49 | 71.74 | 77.87 | 51.71 | 78.07 | 44.00 | 78.23 | 44.98 | 65.76 |
| 4-8-8 | RTN | 74.97 | 71.19 | 72.45 | 42.58 | 67.34 | 40.40 | 74.37 | 43.86 | 60.90 |
| | +QuaRot | 77.97 | 72.06 | 76.16 | 45.48 | 72.31 | 41.40 | 79.82 | 45.34 | 63.82 |
| | +SpinQuant | 79.54 | 71.59 | 78.10 | 48.55 | 72.98 | 42.80 | 75.66 | 45.39 | 64.33 |
| | **+DuaRot** | 79.49 | 73.48 | 78.24 | 50.26 | 78.87 | 44.60 | 80.03 | 46.21 | 66.40 |
| 4-8-16 | RTN | 75.35 | 70.17 | 72.30 | 42.41 | 68.06 | 41.00 | 74.83 | 43.76 | 60.99 |
| | +QuaRot | 77.91 | 71.74 | 76.13 | 45.56 | 72.47 | 41.80 | 79.51 | 45.04 | 63.77 |
| | +SpinQuant | 79.82 | 72.93 | 78.01 | 49.49 | 75.46 | 42.40 | 78.78 | 46.01 | 65.36 |
| | **+DuaRot** | 79.60 | 73.01 | 77.89 | 51.62 | 75.42 | 43.20 | 76.94 | 46.42 | 65.51 |
| 4-4-4 | GPTQ | 52.34 | 48.62 | 25.73 | 22.87 | 27.40 | 30.20 | 48.44 | 33.42 | 36.13 |
| | +QuaRot | 74.65 | 68.11 | 72.82 | 42.24 | 68.48 | 40.80 | 72.17 | 44.37 | 60.46 |
| | +SpinQuant | 77.58 | 68.43 | 75.23 | 47.78 | 75.21 | 42.40 | 76.24 | 44.78 | 63.46 |
| | **+DuaRot** | 78.13 | 69.77 | 76.38 | 47.78 | 75.17 | 43.20 | 78.13 | 44.63 | 64.15 |
| 4-4-8 | GPTQ | 50.76 | 48.54 | 26.75 | 25.85 | 24.62 | 26.80 | 48.47 | 33.32 | 35.64 |
| | +QuaRot | 75.90 | 65.75 | 73.39 | 44.88 | 70.29 | 40.20 | 71.53 | 43.09 | 60.63 |
| | +SpinQuant | 78.35 | 69.77 | 76.06 | 47.44 | 75.46 | 43.00 | 77.92 | 44.73 | 64.09 |
| | **+DuaRot** | 78.29 | 68.98 | 75.55 | 49.06 | 76.18 | 43.40 | 78.17 | 45.34 | 64.37 |
| 4-4-16 | GPTQ | 49.08 | 50.51 | 26.68 | 26.88 | 26.14 | 26.40 | 46.79 | 33.21 | 35.71 |
| | +QuaRot | 76.22 | 69.14 | 73.84 | 43.86 | 70.58 | 40.40 | 71.62 | 42.73 | 61.05 |
| | +SpinQuant | 78.94 | 68.43 | 74.95 | 47.53 | 75.13 | 41.80 | 77.58 | 45.04 | 63.68 |
| | **+DuaRot** | 78.18 | 69.22 | 76.34 | 47.61 | 75.84 | 42.40 | 78.01 | 46.06 | 64.21 |
| 4-8-4 | GPTQ | 54.35 | 52.01 | 38.32 | 24.06 | 36.11 | 27.20 | 53.03 | 34.49 | 39.95 |
| | +QuaRot | 78.94 | 73.72 | 77.22 | 50.85 | 77.48 | 43.80 | 79.08 | 46.57 | 65.96 |
| | +SpinQuant | 79.82 | 73.64 | 77.38 | 51.11 | 77.65 | 43.80 | 80.31 | 45.80 | 66.19 |
| | **+DuaRot** | 80.58 | 71.90 | 77.38 | 50.60 | 75.08 | 44.00 | 78.32 | 45.55 | 65.43 |
| 4-8-8 | GPTQ | 53.97 | 55.72 | 45.51 | 21.42 | 31.82 | 32.20 | 57.31 | 32.19 | 41.27 |
| | +QuaRot | 79.49 | 73.56 | 77.54 | 50.68 | 77.44 | 44.40 | 79.69 | 45.96 | 66.10 |
| | +SpinQuant | 79.98 | 73.48 | 78.01 | 51.62 | 77.78 | 45.40 | 80.18 | 47.19 | 66.71 |
| | **+DuaRot** | 80.36 | 72.77 | 78.09 | 50.60 | 75.76 | 44.20 | 80.43 | 46.06 | 66.03 |
| 4-8-16 | GPTQ | 55.50 | 56.12 | 47.11 | 21.59 | 31.86 | 33.20 | 56.91 | 33.83 | 42.02 |
| | +QuaRot | 79.60 | 73.64 | 77.73 | 50.77 | 77.57 | 44.60 | 79.88 | 46.21 | 66.25 |
| | +SpinQuant | 79.38 | 73.32 | 77.81 | 52.13 | 78.32 | 44.00 | 80.64 | 47.34 | 66.62 |
| | **+DuaRot** | 79.60 | 73.48 | 78.18 | 52.30 | 78.20 | 43.80 | 81.65 | 47.08 | 66.79 |

Table 10: Zero-shot accuracy of Misral-7B with RTN and GPTQ on PIQA, WinoGrande (WG), HellaSwag (HS), ARC-challenge (ARC-c), ARC-easy (ARC-e), OBQA, BoolQ and SIQA.

| W-A-KV | Method | PIQA | WG | HS | ARC-c | ARC-e | OBQA | BoolQ | SIQA | Avg. |
|---|---|---|---|---|---|---|---|---|---|---|
| 16-16-16 | Baseline | 81.88 | 73.95 | 80.98 | 54.95 | 80.18 | 44.40 | 83.49 | 46.83 | 68.33 |
| 4-4-4 | RTN | 53.21 | 49.88 | 26.96 | 25.51 | 26.89 | 23.60 | 39.88 | 34.54 | 35.06 |
| | +QuaRot | 79.33 | 67.96 | 75.79 | 46.76 | 74.12 | 40.60 | 78.26 | 43.76 | 63.32 |
| | +SpinQuant | 79.87 | 70.48 | 78.48 | 48.81 | 76.52 | 42.00 | 82.35 | 44.37 | 65.36 |
| | **+DuaRot** | 81.56 | 70.48 | 78.56 | 50.51 | 78.07 | 41.00 | 80.76 | 43.30 | 65.53 |
| 4-4-8 | RTN | 49.67 | 48.62 | 27.47 | 27.22 | 27.48 | 25.60 | 39.76 | 33.32 | 34.89 |
| | +QuaRot | 77.69 | 68.82 | 75.87 | 46.67 | 75.46 | 39.80 | 78.35 | 43.65 | 63.29 |
| | +SpinQuant | 81.23 | 70.88 | 78.43 | 50.00 | 77.69 | 40.60 | 79.88 | 45.14 | 65.48 |
| | **+DuaRot** | 81.12 | 72.30 | 79.01 | 52.39 | 79.67 | 45.20 | 81.41 | 44.98 | 67.01 |
| 4-4-16 | RTN | 52.45 | 50.59 | 27.57 | 28.07 | 27.86 | 24.20 | 39.42 | 34.34 | 35.56 |
| | +QuaRot | 79.00 | 68.11 | 75.82 | 46.76 | 73.95 | 42.40 | 78.53 | 44.06 | 63.58 |
| | +SpinQuant | 80.41 | 69.14 | 77.74 | 48.04 | 77.10 | 41.40 | 81.90 | 44.98 | 65.09 |
| | **+DuaRot** | 82.15 | 71.67 | 79.35 | 49.15 | 76.43 | 43.20 | 82.48 | 44.47 | 66.11 |
| 4-8-4 | RTN | 78.89 | 66.61 | 76.60 | 48.63 | 73.86 | 40.40 | 70.43 | 44.88 | 62.54 |
| | +QuaRot | 81.12 | 70.96 | 78.77 | 50.94 | 77.78 | 43.40 | 81.25 | 45.09 | 66.16 |
| | +SpinQuant | 81.28 | 72.61 | 79.92 | 52.30 | 79.08 | 43.00 | 82.42 | 46.01 | 67.08 |
| | **+DuaRot** | 81.18 | 72.06 | 79.80 | 54.27 | 79.76 | 44.00 | 81.96 | 45.24 | 67.28 |
| 4-8-8 | RTN | 79.92 | 68.90 | 77.36 | 48.63 | 75.21 | 43.80 | 72.02 | 44.68 | 63.82 |
| | +QuaRot | 81.28 | 72.14 | 78.92 | 50.77 | 78.37 | 44.80 | 81.87 | 44.88 | 66.63 |
| | +SpinQuant | 82.10 | 72.61 | 80.10 | 52.22 | 79.21 | 44.00 | 82.66 | 45.60 | 67.31 |
| | **+DuaRot** | 81.72 | 72.85 | 79.83 | 54.18 | 80.05 | 44.80 | 82.42 | 45.34 | 67.65 |
| 4-8-16 | RTN | 80.09 | 69.14 | 77.56 | 48.89 | 74.75 | 42.00 | 71.62 | 45.45 | 63.69 |
| | +QuaRot | 81.28 | 72.14 | 78.99 | 51.11 | 78.37 | 44.60 | 82.11 | 44.83 | 66.68 |
| | +SpinQuant | 81.66 | 72.30 | 79.80 | 52.05 | 78.58 | 44.40 | 82.42 | 46.42 | 67.20 |
| | **+DuaRot** | 81.94 | 73.01 | 80.20 | 51.71 | 79.12 | 43.40 | 81.68 | 45.50 | 67.07 |
| 4-4-4 | GPTQ | 53.32 | 48.78 | 27.38 | 26.45 | 31.06 | 27.60 | 42.78 | 34.14 | 36.44 |
| | +QuaRot | 79.92 | 68.59 | 78.13 | 50.43 | 76.35 | 41.20 | 80.37 | 45.91 | 65.11 |
| | +SpinQuant | 79.60 | 70.40 | 78.55 | 50.68 | 77.61 | 43.00 | 81.62 | 46.47 | 65.99 |
| | **+DuaRot** | 79.98 | 71.43 | 78.95 | 50.26 | 78.41 | 43.00 | 80.67 | 45.39 | 66.01 |
| 4-4-8 | GPTQ | 53.32 | 48.38 | 27.45 | 25.51 | 30.60 | 26.60 | 44.13 | 36.28 | 36.53 |
| | +QuaRot | 79.05 | 70.24 | 78.07 | 50.09 | 76.52 | 41.40 | 80.21 | 44.17 | 64.97 |
| | +SpinQuant | 79.71 | 71.51 | 78.67 | 51.37 | 78.41 | 43.00 | 81.31 | 44.98 | 66.12 |
| | **+DuaRot** | 80.58 | 72.85 | 78.69 | 51.45 | 78.03 | 43.80 | 81.28 | 46.11 | 66.60 |
| 4-4-16 | GPTQ | 53.97 | 48.54 | 27.55 | 24.74 | 31.73 | 23.40 | 45.47 | 33.73 | 36.14 |
| | +QuaRot | 79.82 | 69.38 | 78.13 | 49.15 | 76.56 | 41.80 | 80.64 | 44.78 | 65.03 |
| | +SpinQuant | 79.87 | 71.67 | 78.83 | 51.19 | 79.00 | 44.60 | 79.97 | 45.96 | 66.39 |
| | **+DuaRot** | 80.30 | 73.16 | 79.02 | 51.28 | 78.87 | 44.00 | 81.96 | 45.85 | 66.81 |
| 4-8-4 | GPTQ | 80.20 | 69.46 | 78.42 | 50.09 | 76.26 | 40.40 | 77.49 | 45.96 | 64.79 |
| | +QuaRot | 81.23 | 73.40 | 79.93 | 53.41 | 79.71 | 42.40 | 82.78 | 45.96 | 67.35 |
| | +SpinQuant | 81.61 | 72.93 | 80.14 | 51.79 | 79.76 | 44.20 | 83.06 | 46.16 | 67.46 |
| | **+DuaRot** | 81.72 | 74.35 | 80.15 | 53.50 | 80.18 | 43.20 | 82.87 | 46.42 | 67.80 |
| 4-8-8 | GPTQ | 80.63 | 72.69 | 79.15 | 50.60 | 78.03 | 42.20 | 78.17 | 46.32 | 65.97 |
| | +QuaRot | 81.45 | 73.95 | 80.05 | 52.82 | 80.22 | 42.60 | 83.03 | 46.37 | 67.56 |
| | +SpinQuant | 81.72 | 73.64 | 80.51 | 53.16 | 80.35 | 45.20 | 82.72 | 46.16 | 67.93 |
| | **+DuaRot** | 80.96 | 74.43 | 80.10 | 52.73 | 80.26 | 43.80 | 82.57 | 45.60 | 67.56 |
| 4-8-16 | GPTQ | 80.63 | 71.27 | 79.34 | 51.02 | 77.74 | 41.80 | 78.23 | 46.37 | 65.80 |
| | +QuaRot | 81.45 | 73.64 | 80.01 | 52.82 | 80.09 | 43.00 | 83.21 | 46.26 | 67.56 |
| | +SpinQuant | 81.72 | 73.32 | 80.42 | 53.24 | 79.84 | 44.60 | 82.84 | 45.91 | 67.74 |
| | **+DuaRot** | 81.23 | 72.85 | 80.22 | 52.99 | 79.76 | 44.60 | 83.73 | 46.47 | 67.73 |

## B  QUANTIZATION ERROR VISUALIZATION

We compare the token-wise quantization errors under different transformations for LLaMA2-7B and LLaMA3-8B. The results are shown below. It can be found that Hadamard, SpinQuant and DuaRot all effectively reduce the quantization errors for tokens, which demonstrates the reason why the rotational invariance of LLM can achieve such a huge improvement compared to the model without rotation. In addition, we can find that, thanks to the dual rotation (global + local), our DuaRot still slightly outperforms SpinQuant in reducing the quantization error.

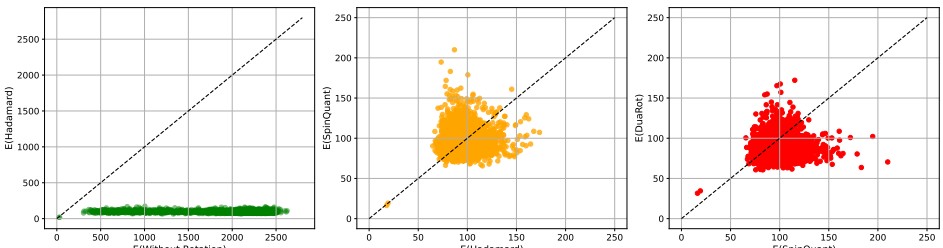

Figure 6: Comparison of token-wise quantization errors without rotation, Hadamard, SpinQuant and DuaRot. Tokens are from LLaMA2-7B model.layers.5.post_attention_layernorm.

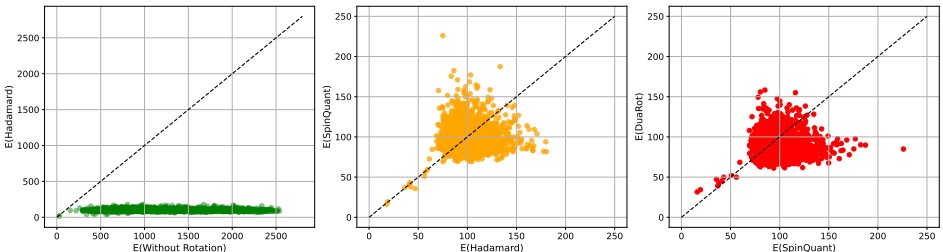

Figure 7: Comparison of token-wise quantization errors without rotation, Hadamard, SpinQuant and DuaRot. Tokens are from LLaMA2-7B model.layers.10.input_layernorm.

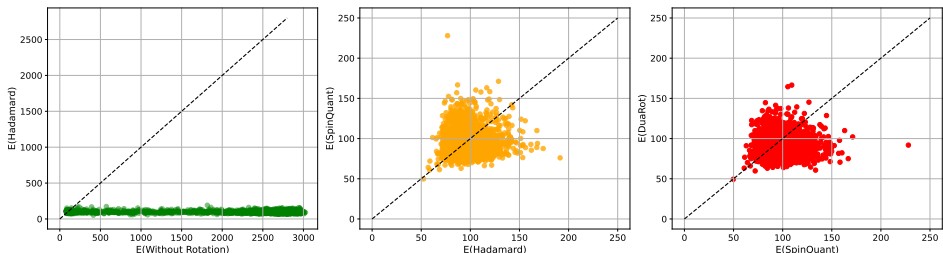

Figure 8: Comparison of token-wise quantization errors without rotation, Hadamard, SpinQuant and DuaRot. Tokens are from LLaMA2-7B model.layers.31.post_attention_layernorm.

## C  PERFORMANCE ANALYSIS

To compare DuaRot with QuaRot and Baseline models, we measure both memory and speed (including Prefill, Decode, and Prefill+Decode) using QuaRot's code [1] on an NVIDIA A100 GPU. From Table 11 and Table 12, we find that addition trainable parameters does not bring significant peak memory usage for LLaMA2-7B. We think this is because LLaMA2-7B has 128 head dim, 32 attention head, 11008 ($172 \times 64$) Ashkboos et al. (2024b) FFN dim, and 32 LlamaDecoderLayer, so the additional parameters brought by our method are:

$$(128 * 128 * 32 + 64 * 64 + 172 * 172) * 32 = 1785497 \approx 0.02B, \tag{12}$$

---

[1] https://github.com/spcl/QuaRot

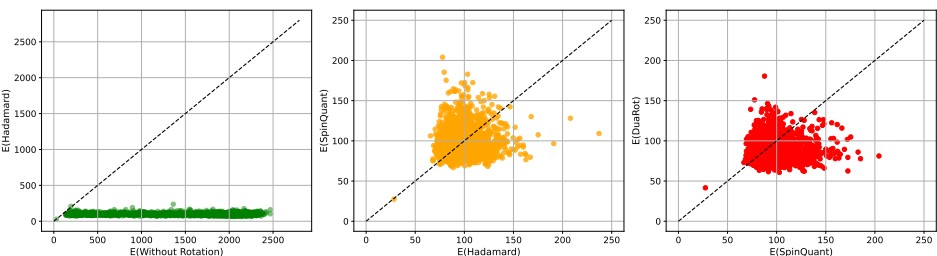

Figure 9: Comparison of token-wise quantization errors without rotation, Hadamard, SpinQuant and DuaRot. Tokens are from LLaMA3-8B model.layers.5.post_attention_layernorm.

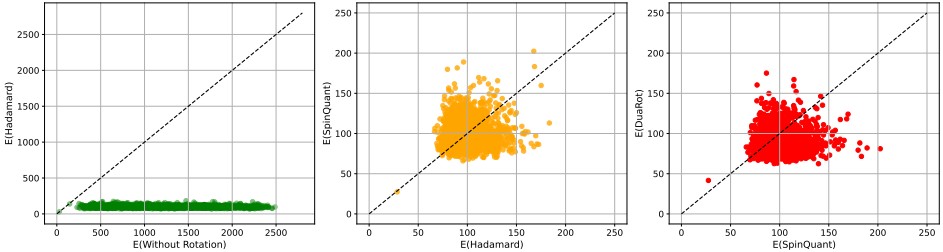

Figure 10: Comparison of token-wise quantization errors without rotation, Hadamard, SpinQuant and DuaRot. Tokens are from LLaMA3-8B model.layers.10.input_layernorm.

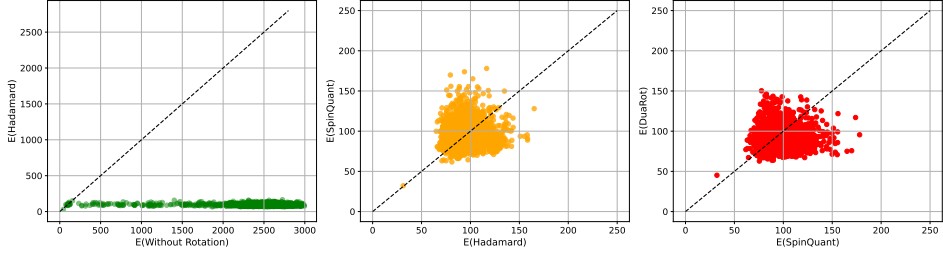

Figure 11: Comparison of token-wise quantization errors without rotation, Hadamard, SpinQuant and DuaRot. Tokens are from LLaMA3-8B model.layers.31.post_attention_layernorm.

Therefore, compared to the LLaMA2-7B model size, it is almost negligible and cannot show a significant difference in peak memory usage.

Meanwhile, as shown in Table 13 and Table 14, although QuaRot always accelerates prefill stage, we can see that Hadamard's WHT always slow down speedup in decode and E2E. This is because, despite the lower computational complexity of WHT, GEMM always has better computational density than WHT on GPUs, and thus tends to correspond to lower latency.

Table 11: Peak Memory usage (in GB) for LLaMA2-7B model with W4A4KV4 quantization strategy on NVIDIA A100. We use 2048 sequence length with different batch sizes. Baseline is FP16 model.

| Model | Batch Size | Sequence Length | Baseline (GB) | QuaRot (GB) | Saving Factor | DuaRot (GB) | Saving Factor |
|---|---|---|---|---|---|---|---|
| LLaMA2-7B | 1 | 2048 | 13.663 | 3.898 | 3.505× | 3.898 | 3.505× |
| | 2 | 2048 | 14.703 | 4.170 | 3.526× | 4.170 | 3.526× |
| | 4 | 2048 | 16.785 | 4.716 | 3.559× | 4.716 | 3.559× |
| | 8 | 2048 | 20.947 | 5.805 | 3.608× | 5.804 | 3.609× |

Table 12: Peak Memory usage (in GB) for LLaMA2-7B model with W4A4KV4 quantization strategy on NVIDIA A100. We use 8 batch sizes with different sequence lengths. Baseline is FP16 model.

| Model | W-A-KV | Batch Size | Sequence Length | Baseline (GB) | QuaRot (GB) | Saving Factor | DuaRot (GB) | Saving Factor |
|-------|--------|------------|-----------------|---------------|-------------|---------------|-------------|---------------|
| LLaMA2-7B | 4-4-4 | 8 | 256 | 13.837 | 3.945 | 3.507 | 3.945 | 3.507 |
| | | 8 | 512 | 14.854 | 4.211 | 3.527× | 4.210 | 3.528× |
| | | 8 | 1024 | 16.885 | 4.742 | 3.561× | 4.742 | 3.561× |
| | | 8 | 2048 | 20.947 | 5.805 | 3.608× | 5.804 | 3.609× |

Table 13: Prefill, Decode, E2E (End to End) speedup for LLaMA2-7B model with W4A4KV4 quantization strategy on NVIDIA A100. We use 50 decode steps with different batch sizes. Baseline is FP16 model.

| Model | Stage | Batch Size | Sequence Length | Baseline (ms) | QuaRot (ms) | Saving Factor | DuaRot (ms) | Saving Factor |
|-------|-------|------------|-----------------|---------------|-------------|---------------|-------------|---------------|
| LLaMA2-7B | Prefill | 1 | 2048 | 258.863 | 220.881 | 1.172× | 206.721 | 1.252× |
| | | 2 | 2048 | 442.280 | 359.857 | 1.229× | 331.118 | 1.336× |
| | | 4 | 2048 | 811.353 | 636.410 | 1.270× | 580.698 | 1.397× |
| | | 8 | 2048 | 1619.521 | 1206.740 | 1.342× | 1089.524 | 1.486× |
| | Decode | 1 | 2048 | 2210.416 | 4331.462 | 0.510× | 3699.385 | 0.598× |
| | | 2 | 2048 | 2192.06 | 4114.290 | 0.533× | 3733.629 | 0.587× |
| | | 4 | 2048 | 2370.041 | 3973.193 | 0.597× | 3910.789 | 0.606× |
| | | 8 | 2048 | 2176.178 | 4290.053 | 0.507× | 3697.386 | 0.589× |
| | E2E | 1 | 2048 | 2459.444 | 4591.451 | 0.536× | 3911.855 | 0.629× |
| | | 2 | 2048 | 2646.515 | 4546.958 | 0.582× | 5120.496 | 0.517× |
| | | 4 | 2048 | 3136.915 | 5539.373 | 0.566× | 4541.015 | 0.691× |
| | | 8 | 2048 | 3725.070 | 6271.752 | 0.594× | 4687.619 | 0.795× |

Table 14: Prefill, Decode, E2E (End to End) speedup for LLaMA2-7B model with W4A4KV4 quantization strategy on NVIDIA A100. We use 50 decode steps with different sequence lengths. Baseline is FP16 model.

| Model | Stage | Batch Size | Sequence Length | Baseline (ms) | QuaRot (ms) | Saving Factor | DuaRot (ms) | Saving Factor |
|-------|-------|------------|-----------------|---------------|-------------|---------------|-------------|---------------|
| LLaMA2-7B | Prefill | 8 | 256 | 202.077 | 170.688 | 1.184× | 153.829 | 1.314× |
| | | 8 | 512 | 388.771 | 304.328 | 1.277× | 281.208 | 1.383× |
| | | 8 | 1024 | 766.327 | 595.146 | 1.288× | 538.952 | 1.422× |
| | | 8 | 2048 | 1619.521 | 1206.740 | 1.342× | 1089.524 | 1.486× |
| | Decode | 8 | 256 | 2255.684 | 4256.577 | 0.530× | 3861.612 | 0.584× |
| | | 8 | 512 | 2275.644 | 4445.924 | 0.512× | 3727.549 | 0.610× |
| | | 8 | 1024 | 2421.642 | 4197.295 | 0.577× | 3808.481 | 0.636× |
| | | 8 | 2048 | 2176.178 | 4290.053 | 0.507× | 3697.386 | 0.589× |
| | E2E | 8 | 256 | 2509.750 | 4293.868 | 0.584× | 4005.369 | 0.627× |
| | | 8 | 512 | 2659.415 | 4514.093 | 0.589× | 4117.514 | 0.646× |
| | | 8 | 1024 | 3151.384 | 4777.762 | 0.660× | 4308.760 | 0.731× |
| | | 8 | 2048 | 3725.070 | 6271.752 | 0.594× | 4687.619 | 0.795× |

# D ADDITION HARDWARE SPEED

Further, we measure the speed on NVIDIA RTX 4090, RTX A6000 and H100-SXM4-80G. As shown in Figure 12, Figure 13 and Figure 14, we can see that Matmul is also faster than WHT when the sequence length is shorter, suggesting that replacing WHT with Matmul can further improve the speed of model inference when the computational density is low.

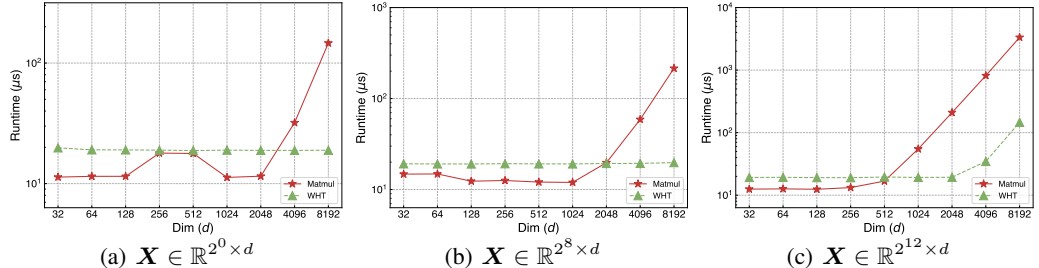

Figure 12: The runtime comparison of the WHT and Matmul for the computation of $\boldsymbol{XH}$ on an NVIDIA RTX 4090 under the different settings of $\boldsymbol{X}$ and $\boldsymbol{H} \in \mathbb{R}^{d \times d}$. We performed computations for $\boldsymbol{XH}$ using torch.float16 and measured the average time over 1000 runs using torch.utils.benchmark.

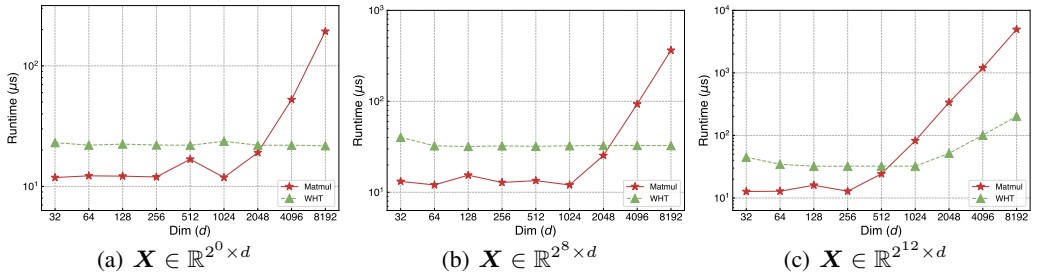

Figure 13: The runtime comparison of the WHT and Matmul for the computation of $\boldsymbol{XH}$ on an NVIDIA RTX A6000 under the different settings of $\boldsymbol{X}$ and $\boldsymbol{H} \in \mathbb{R}^{d \times d}$. We performed computations for $\boldsymbol{XH}$ using torch.float16 and measured the average time over 1000 runs using torch.utils.benchmark.

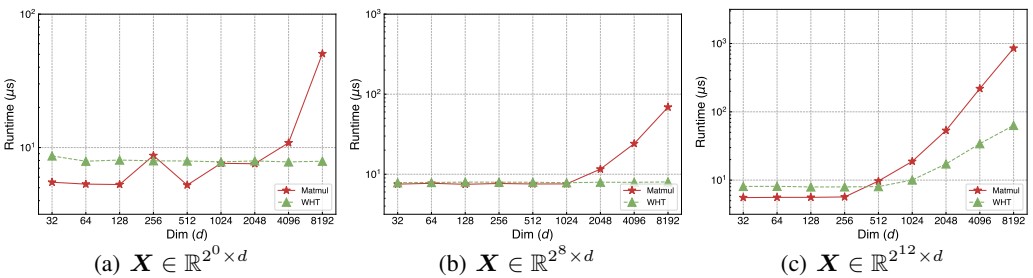

Figure 14: The runtime comparison of the WHT and Matmul for the computation of $\boldsymbol{XH}$ on an NVIDIA H100-SXM4-80GB under the different settings of $\boldsymbol{X}$ and $\boldsymbol{H} \in \mathbb{R}^{d \times d}$. We performed computations for $\boldsymbol{XH}$ using torch.float16 and measured the average time over 1000 runs using torch.utils.benchmark.

Table 15: Comparision between DuaRot (QuaRot, SpinQuant) and DuQuant.

| Method | Rotational Invariance | Deployment | Latency | Accuracy |
|---|---|---|---|---|
| DuaRot (QuaRot, SpinQuant) | Yes | ✓✓ | ✓✓ | ✓ |
| DuQuant | No | ✓ | ✓ | ✓✓ |

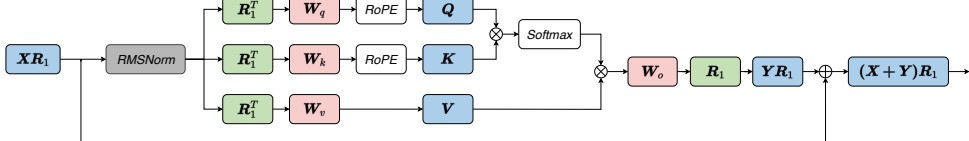

(a) DuaRot (QuaRot, SpinQuant) for Multi-Head Attention (MHA)

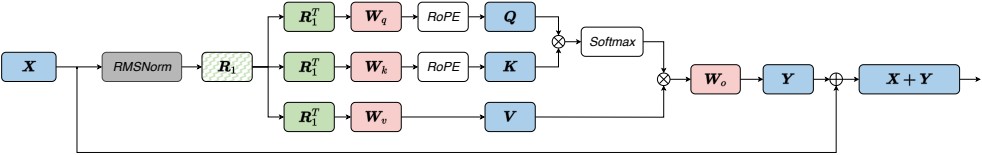

(b) DuQuant for Multi-Head Attention (MHA)

Figure 15: Comparison between DuaRot (QuaRot, SpinQuant) and DuQuant for Multi-Head Attention (MHA).

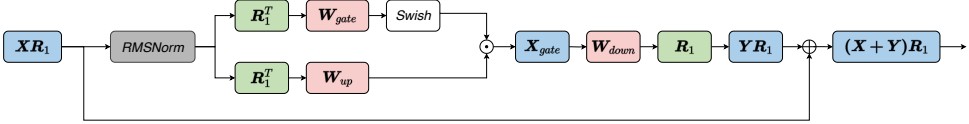

(a) DuaRot (QuaRot, SpinQuant) for Feed-Forward Network (FFN)

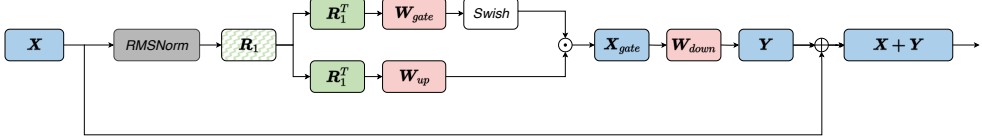

(b) DuQuant for Feed-Forward Network (FFN)

Figure 16: Comparison between DuaRot (QuaRot, SpinQuant) and DuQuant for Feed-Forward Network (FFN).

# E    DUAROT V.S. DUQUANT

We further discuss the essential difference between DuaRot (QuaRot, SpinQuant) and DuQuant in eliminating outliers and massive activation, as a way of explaining why we do not compare DuaRot with DuQuant.

As shown in Figure 15 and Figure 16, DuaRot (QuaRot, SpinQuant) is based on rotational invariance, while DuQuant is not. By applying an equivalent transformation to the network, rotational invariance does not change to the computational graph. which denotes that rotational invariance does not introduce any additional computational cost. However, in Figure 15(b) and Figure 16(b), we can see that DuQuant is not based on rotational invariance. DuQuant inserts $R_1 R_1^T$ in the middle of $XW$ and we can get $X R_1 R_1^T W = X R_1 (R_1^T W)$, although $R_1^T$ can be folded into $W$, $R_1$ must be computed online and will inevitably introduce additional computational cost.

On the other hand, in terms of optimization difficulty, since rotational invariance employs the same $R_1$ throughout the network (including each MHA and FFN), i.e. optimizing $R_1$ will lead to changes in the quantization results of each block, which will lead to the optimization of $R_1$ to be very

difficult. On the contrary, DuQuant uses different $R_1$ for each MHA and FFN, which greatly reduces the optimization difficulty of $R_1$, since the optimization of $R_1$ can be optimized in a Block by Block manner.

As shown in Table 15, from the perspective of ease of deployment, rotational invariance can be seamlessly integrated into existing inference frameworks. However, DuQuant requires modifications to the inference framework and introduces mixed-precision rotation matrix multiplications ($X R_1$). Correspondingly, DuQuant usually can achieve better accuracy. Since the quantization error can be optimized in a block-by-block manner, the quantized model can be effectively improved.

# F    ADDITION ABLATION STUDY

**Global Rotation Matrix v.s. Local Rotation Matrix.**    We conduct ablation studies involving global rotation matrix and local rotation matrix size $d$ on LLaMA3-8B with W4A4KV4 quantization.  The hidden size of LLaMA3-8B is 4096. We select four different settings of $d$, which vary from 512 to 4096 and present the PPL results in Figure 17. As seen, if we only use the local matrix, the model's performance has show a clear positive correlation to the local rotation matrix size. This is because a larger local rotation matrix scatters the outliers over more dimensions, which can achieve better outlier and massive activation elimination.

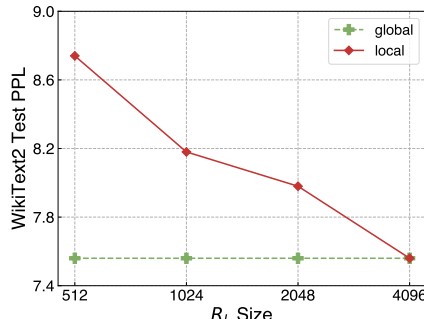

Figure 17: Ablation study on global rotation matrix and local rotation matrix size for W4A4KV4 LLaMA3-8B with RTN.

