# OpenReview forum: "DuaRot: Dual Rotation for Advanced Outlier Mitigation in Rotated LLMs"
_ICLR.cc/2025/Conference — Submitted to ICLR 2025_

### Official Review · Reviewer_5woP · 2024-10-28

**Soundness:** 2
**Presentation:** 2
**Contribution:** 2
**Rating:** 3
**Confidence:** 5

**Summary:**

This paper enhances SpinQuant using two techniques. First, it introduces learnable local rotation, which can be integrated with the original global rotation after training. Additionally, the paper notes that sometimes online Hadamard rotation is slower than same-dimension matrix manipulation. To address this, the paper suggests converting the slower online Hadamard rotation into trainable parameters for more accurate quantization.

**Strengths:**

1. The writing is clear and easy to follow.
2. The paper provides comprehensive experiments and a detailed discussion of related works.

**Weaknesses:**

1. What is the source of the QuaRot and SpinQuant results? Were these results reproduced by the authors? It would be beneficial to report the source of the comparison methods.
2. This method builds on SpinQuant with two new techniques. However, Table 2 shows that the accuracy improvement over SpinQuant is negligible in most cases. While Table 1 shows more significant improvement, it may result from overfitting to the WikiText2 dataset due to more trainable parameters. It would be more reliable to test WikiText2 perplexity on other datasets, such as C4.
3. The online Hadamard rotation matrix is shared across all blocks, while the hardware-aware strategy introduces additional parameters. The paper should discuss the additional parameter overhead.
4. The introduction mentions that "both QuaRot and SpinQuant slow down inference speed during the decoding stage." Does this paper address speeding up the inference during the decoding stage?

**Questions:**

Please refer weakness for details.

---

> ### Author Response · Authors · 2024-11-23
> **Response to Reviewer 5woP**
>
> **Q1**: The source of the QuaRot and SpinQuant results.
>
> **A1**: We reproduced the results based on the source code of QuaRot (https://github.com/spcl/QuaRot) and SpinQuant (https://github.com/facebookresearch/SpinQuant). We have described this in L326 to L328.
>
> **Q2**: It would be more reliable to test WikiText2 perplexity on other datasets, such as C4.
>
> **A2**: In fact, we also find this problem in our experiments. Both SpinQuant and DuaRot suffer from the problem that they can achieve significant improvement on Table 1 but not on Table 2 compared to QuaRot, which will help us to think further in the future about whether there is a better way to measure the quality of quantitative models [1].
>
> In addition, considering the reviewer's comment that “, it may result from overfitting to the WikiText2 dataset”, we also conduct the relevant experiments by using C4 as the calibration set. We discuss it in L477 to L485.
>
> **Q3**: As the hardware-aware strategy introduces additional parameters, the paper should discuss the additional parameter overhead.
>
> **A3**: We add discussion of the additional parameters brought to the paper and provide more complete information in Appendix.C. Thank the reviewer for your valuable suggestions.
>
> **Q4**: Does this paper address speeding up the inference during the decoding stage?
>
> **A4**: We find that one of the main reasons for the speed degradation in the decoding stage is that the computational efficiency of WHT is much lower than that of GEMM in the low-pressure scenario. Based on QuaRot's code, we compare the speed of DuaRot with that of QuaRot at decoding phase in Appendix.C. It can be seen that our hardware-aware based strategy can achieve significant speed gains at decoding stage without introducing significant memory overhead.
>
>
> [1] Hu, Y., Huang, Q., Tao, M., Zhang, C. and Feng, Y., 2024. Can Perplexity Reflect Large Language Model's Ability in Long Text Understanding?. arXiv preprint arXiv:2405.06105.

---

> > ### Comment · Reviewer_5woP · 2024-11-27
> >
> > Thanks for your rebuttal. I have read the rebuttal and decide to maintain my score as 3 (A 4 would be more appropriate, but that's not an option) due to the reported performance come from the serious overfitting problems.
> >
> > Actually, perplexity can reflect the real performance when quantization and test on different dataset. For example, Qserve [1] carry quantization on Pile dataset and report the wikitext2 perplexity. The practical usage of proposed method is limited because most of the performance benefit can be attributed to the over-fitting.
> >
> > [1] Qserve: W4a8kv4 quantization and system co-design for efficient llm serving

---

### Official Review · Reviewer_ssJM · 2024-11-02

**Soundness:** 2
**Presentation:** 3
**Contribution:** 2
**Rating:** 5
**Confidence:** 5

**Summary:**

This paper follows the rotation-based method QuaRot and proposes a method to learn dual rotation matrices for achieving smoother activation distributions.

**Strengths:**

1. The hardware-aware matrix configuration strategy is well-motivated.
2. The illustration of rotational invariance is clear and enhances understanding.

**Weaknesses:**

1. Although the authors mention that SpinQuant and QuaRot rely on GPTQ and aim to achieve smoother activation distributions, DuQuant [1] accomplishes this goal through two orthogonal transformations. There is a lack of discussion and experimental comparison with DuQuant, which directly addresses this motivation.
2. All evaluations are conducted on small-scale language models, leaving the effectiveness of DuaRot on larger models unexplored.
3. There is no measurement of speedup. Given that the authors claim improved efficiency through their matrix configuration strategy, this evalution should be included.
4. The optimization-based approach used to enhance existing baselines is not novel.

[1] DuQuant: Distributing Outliers via Dual Transformation Makes Stronger Quantized LLMs. NeurIPS 2024

**Questions:**

1. Could you discuss and compare DuaRot with DuQuant, which achieves competitive results and a smoother activation landscape without relying on GPTQ?
2. Including figures to illustrate the activation changes might strengthen the argument for DuaRot's effectiveness.
3. Please provide additional evaluation results on the LLaMA3-70B model.
4. Could you include evaluations of memory usage and inference speedup for DuaRot?

---

> ### Author Response · Authors · 2024-11-23
> **Response to Reviewer ssJM**
>
> **Q1**: There is a lack of discussion and experimental comparison with DuQuant, which directly addresses this motivation.
>
> **A1**: We'd be happy to discuss the differences between DuaRot (QuaRot, SpinQuant) and DuQuant based on our understanding in the Appendix. Given the DuQuant pair of computational graph is quite different from the rotational invariance and that this difference leads fundamentally to differences in performance improvement, we do not believe that it would be very appropriate to compare DuaRot (SpinQuant, QuaRot).
>
> **This is a very detailed technique, and we sincerely hope that the reviewer can read our response carefully enough that the reviewer understands rotational invariance to distinguish between the two.**
>
> If the reviewer is in any doubt, we are more than happy to respond further to provide further clarification on this technique.
>
>
> **Q2**: The DuaRot lefts the method’s effectiveness on larger models unexplored.
>
> **A2**: I am very sorry to say that since the exploration of the algorithm on a larger model (e.g. LLaMA3-70B) requies at least 4 $\times$ NVIIDA A100 80GB, it would be too expensive for our lab. As much as we would like to explore, computational resource constraints have prevented us from experimenting. That is why we try our best to conduct experiments on 7B and 13B models. **If you are interested in smaller of models such as meta-llama/Llama-3.2-1B or meta-llama/Llama-3.2-3B, we are more than willing to conduct related experiments in the appendix.**
>
> **Q3**: Speed and memory usage.
>
> **A3**: We use QuaRot's code, and a comparison of memory usage and speed of inference at different stages between DuaRot and QuaRot is presented in the Appendix.C.
>
> **Q4**: The optimization-based approach used to enhance existing baselines is not novel.
>
> **A4**: Yes, our optimization method is also based on Cayley Optimization which is the same to SpinQuant as we also realize that this is a very efficient way to optimize rotation matrices.
>
> The main innovation of our work is that:
>
> 1. We find that although the WHT of Hadamard matrices has lower computational complexity, it does not alway accelerate inference . We can further improve the model accuracy by extending it to the trainable space, and the computational density of GEMM is higher than that of WHT, which does not slow down the inference.
>
> 2. Inspired by RepVGG, we propose a reparameterization method for the rotation matrix, where DuaRot can refine global and local features during training to further eliminate outlier as a way to improve the accuracy of the model. Meanwhile, this reparameterization method can be merged into a single rotation matrix during inference without introducing any additional overhead.
>
> **Q5**: Including figures to illustrate the activation changes.
>
> **A5**: We demonstrate quantization error visualization results in the Appendix.B.

---

> ### Comment · Reviewer_ssJM · 2024-11-25
> **Response to rebuttal**
>
> Thank the authors for their feedback. However, my main concerns have not been addressed, and thus I maintain my score.
>
> - The authors argue against comparing DuaRot with DuQuant by emphasizing differences in rotational invariance.
>   - However, DuQuant applies block-wise rotation, and their experimental results demonstrate **comparable speedup and memory reduction to QuaRot**. Consequently, I do not believe that the rotational invariance is the reason for excluding this work as a baseline.
>   - Adding specific speedup metrics might bolster claims that DuaRot outperforms DuQuant, but **the absence of direct comparative analysis** is a significant oversight.
> - The **unique occurrence of outliers** in the **Weight Matrix of LLaMA3-70B** models are observed in [1].
>   - This is particularly important given the potential impact of such outliers on model performance, where the effectiveness of DuaRot has not yet been explored.
>
> [1] The Uniqueness of LLaMA3-70B Series with Per-Channel Quantization. Arxiv 2024.

---

> ### Author Response · Authors · 2024-11-26
> **Response to Reviewer ssJM**
>
> Thank you very much for the reviewer's suggestion, we are aware of the problem of LLaMA3-70B, but it is very unfortunate that we currently lack sufficient resources to conduct experiments, which is a flaw in our work.
>
> However, with respect to rotational invariance, it is important to emphasize that if we just look at memory and latency, we recognize that DuQuant's scheme doesn't introduce significant overhead compared to QuaRot, but the difference between the two can be understood as:
>
> **We sincerely hope everyone read this can pay their attention**
>
> If we quantize a model to 4bit, **method A** strictly enforces the quantization strategy, while **method B** manually selects some key layers/features to make them unquantized. Although the latter approach achieves significant accuracy gains, introduces no significant overhead and is virtually unsurpassed, we still believe that the study of the former has an undeniable role to play in advancing quantization.
>
> **This is not a very rigorous analogy, but hopefully reviewers will see orthogonal invariance in this sight.**

---

> > ### Comment · Reviewer_ssJM · 2024-11-26
> > **Please Be Polite to Your Reviewer**
> >
> > Thank you for your response. However, I still find some aspects unclear. It seems that QuaRot keeps the Query states at FP16, while DuQuant quantizes the Query states, which makes it difficult to understand the point you're making.
> >
> > Additionally, I would like to kindly request that more respect be given to the reviewers' concerns. I respectfully request a more detailed comparison with DuQuant, as both methods are based on rotation transformation techniques. Moreover, an evaluation of your method on LLaMA3-70B is important, as it is a unique model that could provide valuable insights into the effectiveness of your approach.
> >
> > If you can conduct these experiments and provide the necessary comparisons, I would be happy to reconsider my score. However, based on the current response, I believe a score of 4 is more appropriate.

---

> ### Author Response · Authors · 2024-11-26
>
> Many thanks to the reviewers for their responses, we didn't mean to offend anyone. We just wanted to illustrate the point of view that due to the details in quantization techniques, all critical operations can have a critical impact on the final result. I apologize if this has caused you any offense.
>
> 1. About QKV: Many thanks to the reviewers for their meticulous advice, we really did not realize that DuQuant quantified Query before. We will further explore the impact of Query quantization on model effectiveness in the future. However, with all due respect, according to ICLR review guidelines, DuQuant should be considered as contemporaneous work, and comparison with DuQuant is not mandatory (https://iclr.cc/Conferences/2025/ReviewerGuide)
>
> 2. LLaMA3-70B: We are very sorry that our current computing resources cannot support our experiments on 70B. However, from [1], we are also very confident that exploring 70B is of great interest, and we recognize that this is a shortcoming of our paper.
>
> [1] Qin, M., 2024. The Uniqueness of LLaMA3-70B Series with Per-Channel Quantization. arXiv preprint arXiv:2408.15301.

---

### Official Review · Reviewer_VPVv · 2024-11-03

**Soundness:** 3
**Presentation:** 3
**Contribution:** 2
**Rating:** 5
**Confidence:** 3

**Summary:**

This paper introduces DuaRot, a method that trains global and local rotational matrices independently to effectively mitigate activation outliers.

**Strengths:**

- The motivation for proposing a hardware-aware matrix configuration strategy is strong and well-supported.
- The paper is clearly written and easy to follow.

**Weaknesses:**

- The paper does not provide measurements for speed and memory usage, which I believe are critical evaluations and should be included.
- The experiments are limited to small-scale language models, leaving the method’s effectiveness on larger models unexplored.
- It would be helpful to provide a more direct visualization of the reduction in activation outliers to better illustrate DuaRot’s effectiveness.

**Questions:**

See Weakness

---

> ### Author Response · Authors · 2024-11-23
> **Response to Reviewer VPVv**
>
> **A1**: Speed and memory usage.
>
> **Q1**: We use QuaRot's code, and a comparison of memory usage and speed of inference at different stages between DuaRot and QuaRot is presented in the Appendix.C.
>
> **A2**: The DuaRot lefts the method’s effectiveness on larger models unexplored.
>
> **Q2**: I am very sorry to say that since the exploration of the algorithm on a larger model (e.g. LLaMA3-70B) requies at least 4 $\times$ NVIIDA A100 80GB, it would be too expensive for our lab. As much as we would like to explore, computational resource constraints have prevented us from experimenting. That is why we try our best to conduct experiments on 7B and 13B models. **If you are interested in smaller of models such as meta-llama/Llama-3.2-1B or meta-llama/Llama-3.2-3B, we are more than willing to conduct related experiments in the appendix.**
>
> **A3**: A more direct visualization of the reduction in activation outliers.
>
> **Q3**: We demonstrate quantization error visualization results in the Appendix.B.

---

### Official Review · Reviewer_EjJF · 2024-11-04

**Soundness:** 2
**Presentation:** 3
**Contribution:** 2
**Rating:** 6
**Confidence:** 4

**Summary:**

The paper proposes a rotation-based method to alleviate issues with LLM quantization. The proposed method utilizes two strategies to enhance adaptability and achieves excellent performance even without GPTQ under the INT4 setting.

**Strengths:**

1. The paper is well-structured and developed, making it easy for readers to follow.
2. The motivation for the proposed method is convincing.
3. The method is supported by numerous experiments, which verify its effectiveness from several perspectives.

**Weaknesses:**

1. Some terms mentioned in the paper may be misleading. The phrase "hardware-aware configuration" suggests that the method can automatically adapt to specific hardware. However, it may be more accurately described as a hyperparameter. Additionally, there is results presented for only one type of hardware in the paper.
2. There is a lack of comparison between DuaRot and other baselines in **real runtime** for both training and inference matrices. Since the authors emphasize this point multiple times in the paper, it would be beneficial for them to provide more experimental results in this area.
3. The model size used in the paper is somewhat too small, which limits the persuasiveness of the method's effectiveness.
4. The ablation study is incomplete from my perspective. I believe it would be beneficial to verify the effectiveness of both global and local rotation matrices.

I would consider raising my score if my concerns are addressed.

**Questions:**

1. As mentioned in the Introduction section, the lack of previous work indicates that "both QuaRot and SpinQuant slow down inference speed for the decoding stage." How does DuaRot address this issue? Is there any difference between DuaRot and the mentioned methods in this regard?
2. As noted in Weakness 1, I believe it would be beneficial if you conducted experiments on additional hardware, at least including the 3090 (or 4090). Otherwise, you may want to reconsider the naming of the strategy you used.
3. What do you mean by "w/o DuaRot" in Table 3 and Figure 5? Does it imply that the method degrades to SpinQuant or something else?
4. Could you specify the modules with the online matrix $R^{d\times d}$ where $d \ge 512$ in the LLaMA/Mistral models? If there is a scenario where the online matrix is entirely Hadamard due to the hardware-aware matrix configuration strategy, will this negatively affect performance?
5. It seems peculiar that in Figure 5, when the size of $R_L = 128$, the method shows similar performance with and without DuaRot. Could you please provide a simple explanation for this? Additionally, what do you mean by "instability during training"? Is there any situation where training cannot be completed due to this instability?

---

> ### Author Response · Authors · 2024-11-23
> **Response to Reviewer EjJF**
>
> **Q1**: Additionally hardware results.
>
> **A1**: Based on your suggestion, we have added performance comparisons on the NVIDIA RTX 4090, NVIDIA RTX A6000, and H100-SXM4-80G H100 in Appendix.D.
>
> **Q2**: There is a lack of comparison between DuaRot and other baselines in real runtime for both training and inference matrices.
>
> **A2**:  **Training:** In our experimental setup, the 7/8 B model takes about 25 minutes to train and the 13B takes about 30 minutes to train, which is similar to SpinQuant.
>
> **Inference**: We use QuaRot's code, and a comparison of memory usage and speed of inference at different stages between DuaRot and QuaRot is presented in the Appendix.C.
>
> **Q3**: The DuaRot lefts the method’s effectiveness on larger models unexplored.
>
> **A3**: I am very sorry to say that since the exploration of the algorithm on a larger model (e.g. LLaMA3-70B) requies at least 4 $\times$ NVIIDA A100 80GB, it would be too expensive for our lab. As much as we would like to explore, computational resource constraints have prevented us from experimenting. That is why we try our best to conduct experiments on 7B and 13B models. If you are interested in smaller of models such as meta-llama/Llama-3.2-1B or meta-llama/Llama-3.2-3B, we are more than willing to conduct related experiments in the appendix.
>
> **Q4**: Verifying the effectiveness of both global and local rotation matrices.
>
> **A4**: We demonstrate additional results in Appendix.F.
>
> **Q5**: Discussion on ''both QuaRot and SpinQuant slow down inference speed for the decoding stage''.
>
> **A5**: We acclaim this based on QuaRot issue (https://github.com/spcl/QuaRot/issues/38). We find that one of the main reasons for the decrease in speed is due to the in the decoding phase, where the WHT is much slower than the GEMM, as can be seen in Appendix.C. It is why we propose hardware awareness strategy.
>
> **Q6**: clarification on "w/o DuaRot" in Table 3 and Figure 5.
>
> **A6**: We are very sorry that our description was confusing for you. DuaRot in Table 3 and Figure 5 should be replaced with Dual Rotation. In our method, DuaRot = Dual Rotation + Hardware-awareness. In Figure 5, we used Hard-reness and performed ablation experiments on Dual Rotation.
>
> **Q7**: If there is a scenario where the online matrix is entirely Hadamard due to the hardware-aware matrix configuration strategy, will this negatively affect performance?
>
> **A7**: Given the current scenario, it will hardly exist. We can think about this in two ways:
>
> 1. Only Hadamard matrices of shape $2^n$ can be accelerated using the WHT, therefore:
>
> 2. The larger the model, the shorter the length of the inference using continuous batching. So the sequence length of $2^{12}=4096$ in Figure 4(c) will hardly exist in real scenarios.
>
> **Q8**: It seems peculiar that in Figure 5, when the size of $R_L=128$, the method shows similar performance with and without DuaRot. Could you please provide a simple explanation for this? Additionally, what do you mean by "instability during training"? Is there any situation where training cannot be completed due to this instability?
>
> **A8**: We can understand this in an intuitive way: DuaRot can be viewed as a global rotation followed by a local rotation. If the global rotation has already rotated the features to a relatively smooth situation, then the local rotation may increase the quantization error again, i.e., it may lead to “instability during training”. Larger local rotation matrices are likely to lead to greater instability, so there is still a trade-off in choosing the appropriate parameters.

---

> > ### Comment · Reviewer_EjJF · 2024-12-03
> > **Response**
> >
> > Thanks for the authors' detailed response. Most of my concerns have been adressed, so I have decided to raise my score to 6.

---

### Meta-Review · Area_Chair_XWgj · 2024-12-17

**Metareview:**

The paper presents a dual rotation method aimed at mitigating outlier effects in quantization for large language models (LLMs). This approach seeks to enhance performance without compromising output integrity, marking it as a substantial advancement in the field. Upon review, opinions among the reviewers varied. While one reviewer expressed a borderline positive stance due to the paper's novelty and experimental support, the majority raised significant concerns regarding its limited experimental scope and marginal contributions.
Despite the authors' efforts to address these critiques in their rebuttal, key issues remained unresolved. Concerns regarding the restricted applicability of their findings, based on tests conducted primarily on smaller models, were acknowledged but inadequately mitigated. Notably, problems such as the inadequacy of their experiments to convincingly demonstrate improvements over existing methods, particularly the lack of comparisons with larger models or diverse datasets, persisted. Furthermore, the objection that the reported performance improvements may chiefly result from overfitting has not been satisfactorily addressed.
Ultimately, the consensus among reviewers leans toward recommending rejection due to these unresolved concerns. The promising idea presented holds potential, yet the current execution falls short of the standard required for acceptance. Further work is needed to substantiate the claims and validate the applicability of the proposed method across a broader range of contexts.

**Additional Comments On Reviewer Discussion:**

Despite the authors' efforts to address these critiques in their rebuttal, key issues remained unresolved. Concerns regarding the restricted applicability of their findings, based on tests conducted primarily on smaller models, were acknowledged but inadequately mitigated. Notably, problems such as the inadequacy of their experiments to convincingly demonstrate improvements over existing methods, particularly the lack of comparisons with larger models or diverse datasets, persisted. Furthermore, the objection that the reported performance improvements may chiefly result from overfitting has not been satisfactorily addressed.
Ultimately, the consensus among reviewers leans toward recommending rejection due to these unresolved concerns.

---

### Decision · Program_Chairs · 2025-01-22

Reject